# SLASH: Embracing Probabilistic Circuits into Neural Answer Set Programming

## Abstract

The goal of combining the robustness of neural networks and the expressivity of symbolic methods has rekindled the interest in Neuro-Symbolic AI. Recent advancements in Neuro-Symbolic AI often consider specifically-tailored architectures consisting of disjoint neural and symbolic components, and thus do not exhibit desired gains that can be achieved by integrating them into a unifying framework. We introduce SLASH – a novel deep probabilistic programming language (DPPL). At its core, SLASH consists of Neural-Probabilistic Predicates (NPPs) and logical programs which are united via answer set programming. The probability estimates resulting from NPPs act as the binding element between the logical program and raw input data, thereby allowing SLASH to answer task-dependent logical queries. This allows SLASH to elegantly integrate the symbolic and neural components in a unified framework. We evaluate SLASH on the benchmark data of MNIST addition as well as novel tasks for DPPLs such as missing data prediction and set prediction with state-of-the-art performance, thereby showing the effectiveness and generality of our method.

## 1 Introduction

In recent years, Neuro-Symbolic AI approaches to learning (Hudson & Manning, 2019; d'Avila Garcez et al., 2019; Jiang & Ahn, 2020; d'Avila Garcez & Lamb, 2020), which integrates low-level perception with high-level reasoning by combining data-driven neural modules with logic-based symbolic modules, has gained traction. This combination of sub-symbolic and symbolic systems has been shown to have several advantages for various tasks such as visual question answering and reasoning (Yi et al., 2018), concept learning (Mao et al., 2019) and improved properties for explainable and revisable models (Ciravegna et al., 2020; Stammer et al., 2021).

Rather than designing specifically tailored Neuro-Symbolic architectures, where often the neural and symbolic modules are disjoint and trained independently (Yi et al., 2018; Mao et al., 2019; Stammer et al., 2021), deep probabilistic programming languages (DPPLs) provide an exciting alternative (Bingham et al., 2019; Tran et al., 2017; Manhaeve et al., 2018; Yang et al., 2020). Specifically, DPPLs integrate neural and symbolic modules via a unifying programming framework with probability estimates acting as the *"glue"* between separate modules allowing for reasoning over noisy, uncertain data and, importantly, joint training of the modules. Additionally, prior knowledge and biases in the form of logical rules can easily be added with DPPLs, rather than creating implicit architectural biases, thereby integrating neural networks into downstream logical reasoning tasks.

Object-centric deep learning has recently brought forth several exciting avenues of research by introducing inductive biases to neural networks to extract objects from visual scenes in an unsupervised manner (Zhang et al., 2019; Burgess et al., 2019; Engelcke et al., 2020; Greff et al., 2019; Lin et al., 2020; Locatello et al., 2020; Jiang & Ahn, 2020). We refer to Greff et al. (2020) for a detailed overview. A motivation for this specific line of investigation, which notably has been around for a longer period of time (Fodor & Pylyshyn, 1988; Marcus, 2019), is that objects occur as natural building blocks in human perception and possess advantageous properties for many cognitive tasks, such as scene understanding and reasoning. With a DPPL, these advancements can be improved by integrating the previously mentioned components into the DPPL's programming framework and further adding constraints about objects and their properties in form of logical statements e.g. about color singularity, rather than implicitly enforcing this via one hot encodings.

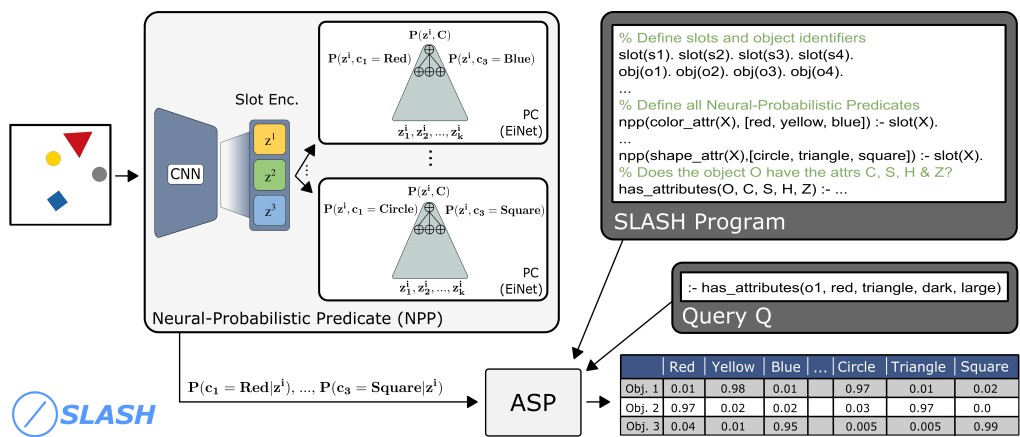

Figure 1: SLASH Attention illustrated for a visual reasoning task. SLASH with Neural-Probabilistic Predicates consisting of a slot attention encoder and Probabilistic Circuits (PCs) realised via EiNets. The slot encoder is shared over all NPPs. Each triangle in the figure represents a single EiNet that gives us a joint distribution at the root node. Thus, each PC learns the joint distribution over slot encodings, $z^i$, and object attributes, $C$, of a specific category, e.g. color attributes. Via targeted queries to the NPPs, one can obtain task-related probabilities, e.g. conditional probabilities for the task of set prediction. Given the probability estimates from the NPP(s) and a SLASH program, containing a set of facts and logical statements about the world, the probability of the truth value of a task-related query are computed via answer set programming. The entire system, including the neural and probabilistic modules, are finally trained end-to-end via a single loss function.

We propose SLASH – a novel DPPL that, similar to the punctuation symbol, can be used to efficiently combine several paradigms into one. Specifically, SLASH represents a scalable programming language that seamlessly integrates probabilistic logical programming with neural representations and tractable probabilistic estimations. Fig. 1 shows an example instantiation of SLASH, termed SLASH Attention, for object-centric set prediction. SLASH consists of several key building blocks. Firstly, it makes use of Neural-Probabilistic Predicates (NPPs) for probability estimation. NPPs consist of neural and/or probabilistic circuit (PC) modules and act as a unifying term, encompassing the neural predicates of DeepProbLog and NeurASP, as well as purely probabilistic predicates. In this work, we introduce a much more powerful *"flavor"* of NPPs that consist jointly of neural and PC modules, taking advantage of the power of neural computations together with true density estimation of PCs. Depending on the underlying task one can thus ask a range of queries to the NPP, e.g. sample an unknown, desired variable, but also query for conditional class probabilities. Example NPPs consisting of a slot attention encoder and several PCs are depicted in Fig. 1 for the task of set prediction. The slot encoder is shared across all NPPs, whereas the PC of each NPP models a separate category of attributes. In this way, each NPP models the joint distribution over slot encodings and object attribute values, such as the color of an object. By querying the NPP, one can obtain task-related probability estimations, such as the conditional attribute probability.

The second component of SLASH is the logical program, which consists of a set of facts and logical statements defining the state of the world of the underlying task. For example, one can define the rules for when an object possesses a specific set of attributes (*cf.* Fig. 1). Thirdly, an ASP module is used to combine the first two components. Given a logical query about the input data, the logical program and the probability estimates obtained from the NPP(s), the ASP module produces a probability estimate about the truth value of the query, stating, e.g., how likely it is for a specific object in an image to be a large, dark red triangle. In contrast to query evaluation in Prolog (Colmerauer & Roussel, 1993; Clocksin & Mellish, 1981) which may lead to an infinite loop, many modern answer set solvers use Conflict-Driven-Clause-Learning (CDPL) which, in principle, always terminates.

Training in SLASH is performed efficiently in a batch-wise and end-to-end fashion, by integrating the parameters of all modules, neural and probabilistic, into a single loss term. SLASH thus allows a simple, fast and effective integration of sub-symbolic and symbolic computations. In our experiments, we investigate the advantages of SLASH in comparison to SOTA DPPLs on the benchmark task of MNIST-Addition (Manhaeve et al., 2018). We hereby show SLASH's increased scalability regarding

computation time, as well as SLASH's ability to handle incomplete data via true probabilistic density modelling. Next, we show that SLASH Attention provides superior results for set prediction in terms of accuracy and generalization abilities compared to a baseline slot attention encoder. With our experiments, we thus show that SLASH is a realization of "one system – two approaches" (Bengio, 2019), that can successfully be used for performing various tasks and on a variety of data types.

We make the following contributions: (1) We introduce neural-probabilistic predicates, efficiently integrating answer set programming with probabilistic inference via our novel DPPL, SLASH. (2) We successfully train neural, probabilistic and logic modules within SLASH for complex data structures end-to-end via a simple, single loss term. (3) We show that SLASH provides various advantages across a variety of tasks and data sets compared to state-of-the-art DPPLs and neural models.

## 2 NEURO-SYMBOLIC LOGIC PROGRAMMING

Neuro-Symbolic AI can be divided into two lines of research, depending on the starting point. Both, however, have the same final goal: to combine low-level perception with logical constraints and reasoning.

A key motivation of Neuro-Symbolic AI (d'Avila Garcez et al., 2009; Mao et al., 2019; Hudson & Manning, 2019; d'Avila Garcez et al., 2019; Jiang & Ahn, 2020; d'Avila Garcez & Lamb, 2020) is to combine the advantages of symbolic and neural representations into a joint system. This is often done in a hybrid approach where a neural network acts as a perception module that interfaces with a symbolic reasoning system, e.g. (Mao et al., 2019; Yi et al., 2018). The goal of such an approach is to mitigate the issues of one type of representation by the other, e.g. using the power of symbolic reasoning systems to handle the generalizability issues of neural networks and on the other hand handle the difficulty of noisy data for symbolic systems via neural networks. Recent work has also shown the advantage of Neuro-Symbolic approaches for explaining and revising incorrect decisions (Ciravegna et al., 2020; Stammer et al., 2021). Many of these previous works, however, train the sub-symbolic and symbolic modules separately.

Deep Probabilistic Programming Languages (DPPLs) are programming languages that combine deep neural networks with probabilistic models and allow a user to express a probabilistic model via a logical program. Similar to Neuro-Symbolic architectures, DPPLs thereby unite the advantages of different paradigms. DPPLs are related to earlier works such as Markov Logic Networks (MLNs) (Richardson & Domingos, 2006). Thereby, the binding link is the Weighted Model Counting (WMC) introduced in $LP^{MLN}$ (Lee & Wang, 2016). Several DPPLs have been proposed by now, among which are Pyro (Bingham et al., 2019), Edward (Tran et al., 2017), DeepProbLog (Manhaeve et al., 2018), and NeurASP (Yang et al., 2020).

To resolve the scalability issues of DeepProbLog, which use Sentential Decision Diagrams (SDDs) (Darwiche, 2011) as the underlying data structure to evaluate queries, NeurASP (Yang et al., 2020), offers a solution by utilizing Answer Set Programming (ASP) (Dimopoulos et al., 1997; Soininen & Niemelä, 1999; Marek & Truszczynski, 1999; Calimeri et al., 2020). In this way, NeurASP changes the paradigm from query evaluation to model generation, i.e. instead of constructing an SDD or similar knowledge representation system, NeurASP generates a set of all possible solutions (one model per solution) and estimates the probability for the truth value of each of these solutions. Of those DPPLs that handle learning in a relational, probabilistic setting and in an end-to-end fashion, all of these are limited to estimating only conditional class probabilities.

## 3 THE SLASH FRAMEWORK

In this section, we introduce our novel DPPL, SLASH. Before we dive into the details of this, it is necessary to first introduce Neural-Probabilistic Predicates, for which we require an understanding of Probabilistic Circuits. Finally, we will present the learning paradigm of SLASH.

The term probabilistic circuit (PC) (Choi et al., 2020) represents a unifying framework that encompasses all computational graphs which encode probability distributions and guarantee tractable probabilistic modelling. These include Sum-Product Networks (SPNs) (Poon & Domingos, 2011) which are deep mixture models represented via a rooted directed acyclic graphs with a recursively defined structure.

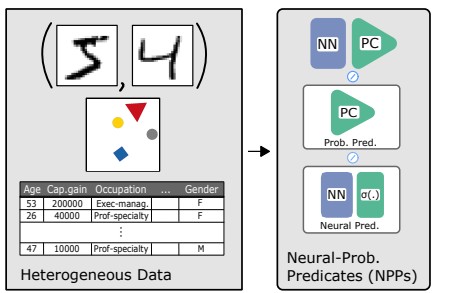

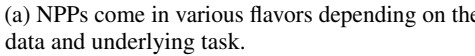

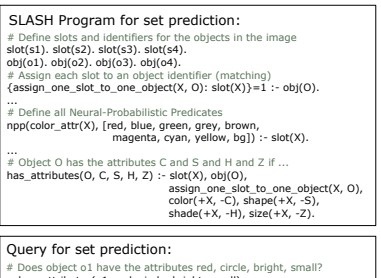

(a) NPPs come in various flavors depending on the data and underlying task.

(b) Minimal SLASH program and query for set prediction.

Figure 2: (a) Depending on the data set and underlying task, SLASH requires a suitable Neural-Probabilistic Predicate (NPP) that computes query-dependent probability estimates. An NPP can be composed of neural and probabilistic modules, or (depicted via slash symbol) only one of these two. (b) A minimal SLASH program and query for the set prediction task, here only showing the NPP that models the color category per object. For the full program, we refer to the Appendix.

### 3.1 NEURAL-PROBABILISTIC PREDICATES

Previous DPPLs, DeepProbLog (Manhaeve et al., 2018) and NeurASP (Yang et al., 2020), introduced the *Neural Predicate* as an annotated-disjunction or as a propositional atom, respectively, to acquire conditional class probabilities, $P(C|X)$, via the softmax function at the output of an arbitrary DNN. As mentioned in the introduction, this approach has certain limitations concerning inference capabilities. To resolve this issue, we introduce *Neural-Probabilisitic Predicates* (NPPs).

Formally, we denote with

$$npp\left(h(x), [v_1, \ldots, v_n]\right) \tag{1}$$

a Neural-Probabilistic Predicate $h$. Thereby, (i) *npp* is a reserved word to label an NPP, (ii) $h$ a symbolic name of either a PC, NN or a joint of a PC and NN (*cf.* Fig. 2a), e.g., *color_attr* is the name of an NPP of Fig. 2b. Additionally, (iii) $x$ denotes a "term" and (iv) $v_1, \ldots, v_n$ are placeholders for each of the $n$ possible outcomes of $h$. For example, the placeholders for *color_attr* are the color attributes of an object (*Red, Blue, Green, etc.*).
An NPP abbreviates an arithmetic literal of the form $c = v$ with $c \in \{h(x)\}$ and $v \in \{v_1, \ldots, v_n\}$. Furthermore, we denote with $\Pi^{npp}$ a set of NPPs of the form stated in (Eq. 1) and $r^{npp}$ the set of all rules $c = v$ of one NPP, which denotes the possible outcomes, obtained from an NPP in $\Pi^{npp}$, e.g. $r^{color\_attr} = \{c = Red, c = Blue, c = Green, ...\}$ for the example depicted in Fig. 2b.

Rules of the form $npp\left(h(x), [v_1, \ldots, v_n]\right) \leftarrow Body$ are used as an abbreviation for application to multiple entities, e.g. multiple slots for the task of set prediction (*cf.* Fig. 2b). Hereby, *Body* of the rule is identified by $\top$ (tautology, true) or $\bot$ (contradiction, false) during grounding. Rules of the form *Head* $\leftarrow$ *Body* with $r^{npp}$ appearing in *Head* are prohibited for $\Pi^{npp}$.

In this work, we largely make use of NPPs that contain probabilistic circuits (specifically SPNs) which allow for tractable density estimation and modelling of joint probabilities. In this way, it is possible to answer a much richer set of probabilistic queries, i.e. $P(X, C)$, $P(X|C)$ and $P(C|X)$.

In addition to this, we introduce the arguably more interesting type of NPP that combines a neural module with a PC. Hereby, the neural module learns to map the raw input data into an optimal latent representation, e.g. object-based slot representations. The PC, in turn, learns to model the joint distribution of these latent variables and produces the final probability estimates. This type of NPP nicely combines the representational power of neural networks with the advantages of PCs in probability estimation and query flexibility.

For making the different probabilistic queries distinguishable in a SLASH program, we introduce the following notation. We denote a given variable with $+$ and the query variable with $-$. E.g., within the running example of set prediction (*cf.* Fig. 1 and 2b), with the query $color\_attr(+X, -C)$ one is asking for $P(C|X)$. Similarly, with $color\_attr(-X, +C)$ one is asking for $P(X|C)$ and, finally, with $color\_attr(-X, -C)$ for $P(X, C)$.

To summarize, an NPP can consist of neural and/or probabilistic modules and produces query-dependent probability estimates. Due to the flexibility of its definition, the term NPP contains the predicates of previous works (Manhaeve et al., 2018; Yang et al., 2020), but also more interesting predicates discussed above. The specific "flavor" of an NPP should be chosen depending on what type of probability estimation is required (*cf.* Fig 2a). Lastly, NPPs have the unified loss function of the negative log-likelihood:

$$L_{NPP} := -\log LH(x, \hat{x}) = \sum_{i=1}^{n} LH(x_i, \hat{x}_i) = -\sum_{i=1}^{n} x_i \cdot \log(P_\xi^{(X,C)}(x_i)) = -\sum_{i=1}^{n} \log(P_\xi^{(X,C)}) \tag{2}$$

whereby we are assuming the data to be i.i.d., ground truth $x_i$ to be the all-ones vector, $\xi$ to be the parameters of the NPP and $P_\xi^{(X,C)}$ are the predictions $\hat{x}_i$ obtained from the PC encoded in the NPP.

### 3.2 THE SLASH LANGUAGE AND SEMANTICS

Fig. 1 presents an illustration of SLASH, exemplified for the task of set prediction, with all of its key components. Having introduced the NPPs previously, which produce probability estimates, we now continue in the pipeline on how to use these probability estimates for answering logical queries. We begin by formally defining a SLASH program.

**Definition 1.** *A SLASH program $\Pi$ is the union of $\Pi^{asp}$, $\Pi^{npp}$. Therewith, $\Pi^{asp}$ is the set of propositional rules (standard rules from ASP-Core-2 (Calimeri et al., 2020)), and $\Pi^{npp}$ is a set of Neural-Probabilistic Predicates of the form stated in Eq. 1.*

Fig. 2b depicts a minimal SLASH program for the task of set prediction, exemplifying a set of propositional rules and neural predicates. Similar to NeurASP, SLASH requires ASP and as such adopts its **syntax** to most part. We therefore now address integrating our NPPs into an ASP compatible form to obtain the success probability for the logical query given all possible solutions. Thus, we define **SLASH's semantics**. For SLASH to translate the program $\Pi$ to the ASP-solver's compatible form, the rules (Eq. 1) will be rewritten to the set of rules:

$$1\{h(x) = v_1; \ldots; h(x) = v_n\}1 \tag{3}$$

The ASP-solver should understand this as "Pick exactly one rule from the set". After the translation is done, we can ask an ASP-solver for the solutions for $\Pi$. We denote a set of ASP constraints in the form $\leftarrow Body$, as *queries $Q$* (annotation). and each of the solutions with respect to $Q$ as a *potential solution*, $I$, (referred to as *stable model* in ASP). With $I|_{r^{npp}}$ we denote the projection of the $I$ onto $r^{npp}$, $Num(I|_{r^{npp}}, \Pi)$ – the number of the possible solutions of the program $\Pi$ agreeing with $I|_{r^{npp}}$ on $r^{npp}$. Because we aim to calculate the success probability of the query $Q$, we formalize the probability of a potential solution $I$ beforehand.

**Definition 2.** *We specify the probability of the potential solution, $I$, for the program $\Pi$ as the product of the probabilities of all atoms $c = v$ in $I|_{r^{npp}}$ divided by the number of potential solutions of $\Pi$ agreeing with $I|_{r^{npp}}$ on $r^{npp}$:*

$$P_\Pi(I) = \begin{cases} \frac{\prod_{c=v \in I|_{r^{npp}}} P_\Pi(c=v)}{Num(I|_{r^{npp}}, \Pi)}, & \text{if } I \text{ is a potential solution of } \Pi, \\ 0, & \text{otherwise.} \end{cases} \tag{4}$$

Therefore, the probability of a query can be defined as follows.

**Definition 3.** *The probability of the query $Q$ given the set of possible solutions $I$ is defined as*

$$P_\Pi(Q) := \sum_{I \models Q} P_\Pi(I). \tag{5}$$

*Thereby, $I \models Q$ reads as "I satisfies $Q$". The probability of the set of queries $\mathbf{Q} = \{Q_1, \ldots, Q_l\}$ is defined as the product of the probability of each. I.e.*

$$P_\Pi(\mathbf{Q}) := \prod_{Q_i \in \mathbf{Q}} P_\Pi(Q_i) = \prod_{Q_i \in \mathbf{Q}} \sum_{I \models Q} P_\Pi(I). \tag{6}$$

### 3.3 PARAMETER LEARNING IN SLASH

We denote with $\Pi(\boldsymbol{\theta})$ the SLASH program under consideration, thereby $\boldsymbol{\theta}$ is the set of the parameters associated with $\Pi$. Further, making the i.i.d. assumption of the query set $\mathbf{Q}$, we follow Manhaeve et al. (2018) and Skryagin et al. (2020), and use the *learning from entailment* setting. That is, the training examples are logical queries that are known to be true in the SLASH program $\Pi(\boldsymbol{\theta})$. The goal is now to learn the parameters $\boldsymbol{\theta}$ of the SLASH program $\Pi(\boldsymbol{\theta})$ so that the observed queries are most likely.

To this end, we employ the negative log-likelihood and the cross-entropy of the observed queries $P_{\Pi(\theta)}(Q_i)$ and their predicted probability value $P^{(X_\mathbf{Q}, C)}(x_{Q_i})$, assuming the NPPs are fixed: $L_{ENT} :=$

$$-\log LH\left(\log(P_{\Pi(\theta)}(\mathbf{Q})), P^{(X_\mathbf{Q}, C)}(x_\mathbf{Q})\right) = -\sum_{j=1}^{m} \log(P_{\Pi(\theta)}(Q_{ij})) \cdot \log\left(P^{(X_\mathbf{Q}, C)}(x_{Q_{ij}})\right). \quad (7)$$

This loss function aims at maximizing the estimated success probability. We remark that the defined loss function is true regardless of the NPP's form (NN with Softmax, PC or PC jointly with NN). The only difference will be the second term, i.e. $P^{(C|X_\mathbf{Q})}(x_\mathbf{Q})$ or $P^{(X_\mathbf{Q}|C)}(x_\mathbf{Q})$) depending on the NPP and task. Furthermore, we assume that for the set of queries $\mathbf{Q}$ holds

$$P_{\Pi(\boldsymbol{\theta})}(Q) > 0 \quad \forall Q \in \mathbf{Q}.$$

In accordance with the semantics, we seek to reward the right solutions $v = c$ and penalize wrong ones $v \neq c$. Referring to the probabilities in $r^{npp}$ (the set of logical rules denoting NPPs, see Def. 2) as $\mathbf{p}$, one can compute their gradients w.r.t. $\boldsymbol{\theta}$ via backpropagation as

$$\sum_{Q \in \mathbf{Q}} \frac{\partial \log\left(P_{\Pi(\boldsymbol{\theta})}(Q)\right)}{\partial \boldsymbol{\theta}} = \sum_{Q \in \mathbf{Q}} \frac{\partial \log\left(P_{\Pi(\boldsymbol{\theta})}(Q)\right)}{\partial \mathbf{p}} \times \frac{\partial \mathbf{p}}{\partial \boldsymbol{\theta}} . \quad (8)$$

The term $\frac{\partial \mathbf{p}}{\partial \boldsymbol{\theta}}$ can now be computed as usual via backward propagation through the NPPs (see Eq. 13 in the appendix for details). By letting $p$ to be the label of the probability of an atom $c = v$ in $r^{npp}$ and denoting $P_{\Pi(\boldsymbol{\theta})}(c = v)$, the term $\frac{\partial \log\left(P_{\Pi(\boldsymbol{\theta})}(Q)\right)}{\partial \mathbf{p}}$ follows from NeurASP (Yang et al., 2020) as

$$\frac{\partial \log\left(P_{\Pi(\boldsymbol{\theta})}(Q)\right)}{\partial \mathbf{p}} = \frac{\displaystyle\sum_{\substack{I:I \models Q \\ I \models c = v}} \frac{P_{\Pi(\boldsymbol{\theta})}(I)}{P_{\Pi(\boldsymbol{\theta})}(c = v)} - \sum_{\substack{I:I, v' \models Q \\ I \models c = v', v \neq v'}} \frac{P_{\Pi(\boldsymbol{\theta})}(I)}{P_{\Pi(\boldsymbol{\theta})}(c = v')}}{\displaystyle\sum_{I:I \models Q} P_{\Pi(\boldsymbol{\theta})}(I)} . \quad (9)$$

This is sensible. For instance, if a query to be true is not likely to be entailed, the gradient is positive. Putting everything together, the final loss function is

$$L_{SLASH} = L_{NPP} + L_{ENT} \quad (10)$$

and we perform training using coordinate descent, i.e., we train the NPPs, the train the program with fixed NPPs, train the NPPs with the program fixed, and so on.

In hindsight, rather than requiring a novel loss function for each individual task and data set, with SLASH, it is possible to simply incorporate the specific requirements into the logic program. The training loss, however, remains the same. We refer to the Appendix A for further details, including the derivation of the total loss gradient.

## 4 EMPIRICAL RESULTS

The advantage of SLASH lies in the efficient integration of neural, probabilistic and symbolic computations. To emphasize this, we conduct a variety of experimental evaluations.

**Experimental Details.** We use two benchmark data sets, namely MNIST (LeCun et al., 1998b) for the task of MNIST-Addition and a variant of the ShapeWorld data set (Kuhnle & Copestake, 2017) for

Table 1: MNIST Addition Results. Test accuracy corresponds to the percentage of correctly classified test images. (a) Test accuracies in percent for the MNIST Addition task with various DPPLs, including SLASH with an NPP that models the joint probabilities (SLASH (PC)) and one that models only conditional probabilities (SLASH (DNN)). (b) Test accuracies in percent for the MNIST Addition task with missing data, comparing DeepProbLog with SLASH (PC). The amount of missing data was varied between 50% and 97% of the pixels per image.

(a) Baseline MNIST Addition.

|  | Test Acc. (%) |
|---|---|
| DeepProbLog | $98.49 \pm 0.18$ |
| NeurASP | $98.21 \pm 0.30$ |
| **SLASH** (PC) | $95.39 \pm 0.29$ |
| **SLASH** (DNN) | $\mathbf{98.74 \pm 0.21}$ |

(b) Missing data MNIST Addition.

|  | **DeepProbLog** | **SLASH** (PC) |
|---|---|---|
| 50% | $\mathbf{97.73 \pm 0.12}$ | $97.67 \pm 0.12$ |
| 80% | $76.07 \pm 18.38$ | $\mathbf{96.72 \pm 0.05}$ |
| 90% | $69.15 \pm 29.15$ | $\mathbf{94.85 \pm 0.38}$ |
| 97% | $32.46 \pm 22.48$ | $\mathbf{82.57 \pm 4.66}$ |

object-centric set prediction. For all experiments we present the average and the standard deviation over five runs with different random seeds for parameter initialization.

For ShapeWorld experiments, we generate a data set we refer to as ShapeWorld4. Images of ShapeWorld4 contain between one and four objects, with each object consisting of four attributes: a color (red, blue, green, gray, brown, magenta, cyan or yellow), a shade (bright, or dark), a shape (circle, triangle or square) and a size (small or big). Thus, each object can be created from 84 different combinations of attributes. Fig. 1 depicts an example image.

We measure performance via classification accuracies in the MNIST-Addition task. In our Shape-World4 experiments, we present the average precision. We refer to appendix B for the SLASH programs and queries of each experiment, and appendix C for a detailed description of hyperparameters and further details.

**Evaluation 1: SLASH outperforms SOTA DPPLs in MNIST-Addition.** The task of MNIST-Addition (Manhaeve et al., 2018) is to predict the sum of two MNIST digits, presented only as raw images. During test time, however, a model should classify the images directly. Thus, although a model does not receive explicit information about the depicted digits, it must learn to identify digits via indirect feedback on the sum prediction.

We compare the test accuracy after convergence between the three DPPLs: DeepProbLog (Manhaeve et al., 2018), NeurASP (Yang et al., 2020) and SLASH, using a probabilistic circuit (PC) or a deep neural network (DNN) as NPP. Notably, the DNN used in SLASH (DNN) is the LeNet5 model (LeCun et al., 1998a) of DeepProbLog and NeurASP. We note that when using the PC as NPP, we have also extracted conditional class probabilities $P(C|X)$, by marginalizing the class variables $C$ to acquire the normalization constant $P(X)$ from the joint $P(X, C)$, and calculating $P(X|C)$.

The results can be seen in Tab. 1a. We observe that training SLASH with a DNN NPP produces SOTA accuracies compared to DeepProbLog and NeurASP, confirming that SLASH's batch-wise loss computation leads to improved performances. We further observe that the test accuracy of SLASH with a PC NPP is slightly below the other DPPLs, however we argue that this may be since a PC, in comparison to a DNN, is learning a true mixture density rather than just conditional probabilities. The advantages of doing so will be investigated in the next experiments. Note that, optimal architecture search for PCs, e.g. for computer vision, is an open research question.These evaluations show SLASH's advantages on the benchmark MNIST-Addition task. Additional benefits will be made clear in the following experiments.

**Evaluation 2: Handling Missing Data with SLASH.** SLASH offers the advantage of its flexibility to use various kinds of NPPs. Thus, in comparison to previous DPPLs, one can easily integrate NPPs into SLASH that perform joint probability estimation. For this evaluation, we consider the task of MNIST-Addition with missing data. We trained SLASH (PC) and DeepProbLog with the MNIST-Addition task with images in which a percentage of pixels per image has been removed. It is important to mention here that whereas DeepProbLog handles the missing data simply as background pixels, SLASH (PC) specifically models the missing data as uncertain data by marginalizing the denoted pixels at inference time. We use DeepProbLog here representative of DPPLs without true density estimation.

(a) ShapeWorld4 and ShapeWorld4 CoGenT Test Avg. Precision

|  | Slot Att. | **SLASH Att.** |
|---|---|---|
| **ShapeWorld4** | | |
| Test Set | $90.24 \pm 0.93$ | $\mathbf{95.58 \pm 0.61}$ |
| **CoGenT** | | |
| Test Cond. A | $90.37 \pm 2.19$ | $\mathbf{96.85 \pm 0.43}$ |
| Test Cond. B | $27.15 \pm 2.36$ | $\mathbf{40.58 \pm 1.99}$ |

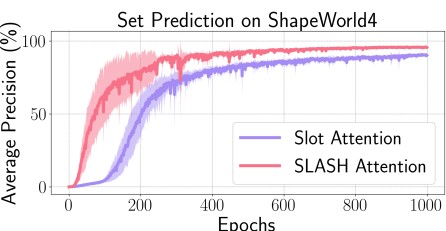

(b) Test Avg. Precision over Training Epochs

Figure 3: ShapeWorld4 Experiments. (a) Converged test average precision scores for the set prediction task with ShapeWorld4 (top) and ShapeWorld4 CoGenT (bottom). (b) Test average precision scores for set prediction with ShapeWorld4 over the training epochs. In these experiments we compared a baseline slot encoder versus SLASH Attention with slot attention and PC-based NPPs. For the CoGenT experiments, a model is trained on one training set and tested on two separate test conditions. The Condition A test set contains attribute compositions which were also seen during training. The Condition B test set contains attribute compositions which were not seen during training, e.g. yellow circles were not present in the training set, but present in Condition B test set.

The results can be seen in Tab. 1b for $50\%$, $80\%$, $90\%$ and $97\%$ missing pixels per image. We observe that at $50\%$, DeepProbLog and SLASH produce almost equal accuracies. With $80\%$ percent missing pixels, there is a substantial difference in the ability of the two DPPLs to correctly classify images, with SLASH being very stable. By further increasing the percentage of missing pixels, this difference becomes even more substantial with SLASH still reaching a $82\%$ test accuracy even when $97\%$ of the pixels per image are missing, whereas DeepProbLog degrades to an average of $32\%$ test accuracy. We further note that SLASH, in comparison to DeepProbLog, produces largely reduced standard deviations over runs.

Thus, by utilizing the power of true density estimation SLASH, with an appropriate NPP, can produce more robust results in comparison to other DPPLs. Further, we refer to Appendix D, which contains results of additional experiments where training is performed with the full MNIST data set whereas only the test set entails different rates of missing pixels.

**Evaluation 3: Improved Concept Learning via SLASH.** We show that SLASH can be very effective for the complex task of set prediction, which previous DPPLs have not tackled. We revert to the ShapeWorld4 data set for this setting. For set prediction, a model is trained to predict the discrete attributes of a set of objects in an image (*cf.* Fig. 1 for an example ShapeWorld4 image). The difficulty for the model lies therein that it must match an unordered set of corresponding attributes (with varying number of entities over samples) with its internal representations of the image.

The slot attention module introduced by Locatello et al. (2020) allows for an attractive object-centric approach to this task. Specifically, this module represents a pluggable, differentiable module that can be easily added to any architecture and, through a competitive softmax-based attention mechanism, can enforce the binding of specific parts of a latent representation into permutation-invariant, task-specific vectors, called slots.

In our experiments, we wish to show that by adding logical constraints to the training setting, one can improve the overall performances and generalization properties of such a model. For this, we train SLASH with NPPs as depicted in Fig. 1 consisting of a shared slot encoder and separate PCs, each modelling the mixture of latent slot variables and the attributes of one category, e.g. color. For ShapeWorld4, we thereby have altogether four NPPs. SLASH is trained via queries of the kind exemplified in Fig. 7 in the Appendix. We refer to this configuration as *SLASH Attention*.

We compare SLASH Attention to a baseline slot attention encoder using an MLP and Hungarian loss for predicting the object properties from the slot encodings as in Locatello et al. (2020). The results of these experiments can be found in Fig. 3a (top). We observe that the average precision after convergence on the held-out test set with SLASH Attention is greatly improved to that of the baseline model. Additionally, in Fig. 3b we observe that SLASH Attention reaches the average precision value of the baseline model in much fewer number of epochs. Thus, we can summarize that adding logical knowledge in the training procedure via SLASH can greatly improve the capabilities of a neural module for set prediction.

**Evaluation 4: Improved Compositional Generalization with SLASH.** To test the hypothesis that SLASH Attention possesses improved generalization properties in comparison to the baseline model, we ran experiments on a variant of ShapeWorld4 similar to the CLEVR Compositional Generalization Test (CoGenT) (Johnson et al., 2017). The goal of CoGenT is to investigate a model's ability to handle novel combinations of attributes that were not seen during training.

For this purpose, we established two conditions within a ShapeWorld4 CoGenT data set: **Condition (A)** – the training and test data set contains squares with the colors *gray*, *blue*, *brown*, or *yellow*, triangles with the colors *red*, *green*, *magenta*, or *cyan* and circles of *all* colors. **Condition (B)** – the training set is as in Condition (A). However, the test set contains squares with the colors *red*, *green*, *magenta*, or *cyan*, triangles with the colors *gray*, *blue*, *brown*, or *yellow* and circles of *all* colors. The goal is to investigate how well a model can generalize that, e.g., also squares can have the color red, although never having seen evidence for this during training.

The resulting average precision test scores are presented in Fig. 3a (bottom). We observe that, even though the SLASH Program used for this experiment was not explicitly written to handle composition generalization, SLASH Attention shows greatly improved generalization capabilities. This can be seen in the approx. 13% higher average precision scores on the Condition (B) test set in comparison to the baseline model. Importantly, this trend still holds even when subtracting the higher precision scores observed in Condition (A).

To summarize our findings from the experiments on set prediction: we observe that adding prior knowledge in the form of logical constraints via SLASH can greatly improve a neural module in terms of performance and generalizability. On a side note: training neural networks for novel tasks, often involves defining explicit loss functions, e.g. Hungarian loss for set prediction. In contrast with SLASH, no matter the choice of NPP and underlying task, the training loss remains the same. Task-related requirements simply need to be added as lines of code to the SLASH program. This additionally highlights SLASH's versatility and flexibility.

**Summary of all Empirical Results.** All empirical results together demonstrate that the flexibility of SLASH is highly beneficial and can easily outperform state-of-the-art: one can freely combine what is required to solve the underlying task — (deep) neural networks, PCs, and logic. Particularly, the results indicate the potential of integrating PCs via SLASH into DPPLs.

# 5 Conclusion and Future Work

We introduce SLASH, a novel DPPL that integrates neural computations with tractable probability estimates and logical statements. The key ingredient of SLASH to achieve this are Neural-Probabilistic Predicates (NPPs) that can be flexibly constructed out of neural and/or probabilistic circuit modules based on the data and underlying task. With these NPPs, one can produce task-specific probability estimates. The details and additional prior knowledge of a task are neatly encompassed within a SLASH program with only few lines of code. Finally, via Answer Set Programming and Weighted Model Counting, the logical SLASH program and probability estimates from the NPPs are combined to estimate the truth value of a task-specific query. Our experiments show the power and efficiency of SLASH, improving upon previous DPPLs in the benchmark MNIST-Addition task in terms of performance, efficiency and robustness. Importantly, by integrating a SOTA slot attention encoder into NPPs and adding few logical constraints, SLASH demonstrates improved performances and generalizability in comparison to the pure slot encoder for the task of object-centric set prediction; a setting no DPPL has tackled yet. This shows the great potential of DPPLs to elegantly combine logical reasoning with neural computations and uncertainty estimates.

Interesting avenues for future work include benchmarking SLASH on additional data types and tasks. One should explore unsupervised and weakly supervised learning using logic with SLASH and investigate how far logical constraints can help unsupervised object discovery. In direct alignment with our work, one should also investigate image generation via the beneficial feature of PCs to generate random samples. Actually, it should be possible to generate images that encapsulate logical knowledge bases. This is important to move from data-rich to knowledge-rich AI.

## ETHICS STATEMENT

With our work, we have shown that one can add prior knowledge and logical constraints to the training of learning systems. We postulate that SLASH can therefore additionally be used to identify and remove biases or undesirable behavior, by adding constraints within the SLASH program. We observe that this feature, however, also has the potential danger to be used in the opposite way, e.g. explicitly adding bias and discriminatory factors to a system. To the best of our knowledge, our study does not raise any ethical, privacy or conflict of interest concerns.

## REPRODUCIBILITY STATEMENT

An official, curated GitHub repository will be made public with the final version, containing the code of SLASH, as well as scripts to reproduce the experiments and generate data sets. In addition to this, architectural details and hyperparameters are included in the appendix. Preliminary code will be uploaded upon submission. Lastly, details on the evaluation metrics and relevant data sets are given in the main text as well as appendix.

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

## A   APPENDIX A – DETAILS ON PARAMETER LEARNING

In the Appendix, we want to discuss details on **parameter learning** in SLASH. Since we use co-ordinate descent for training SLASH we present the derivative of each component of the loss function defined in equation 10 since while optimization, one component has to be kept fixed.

We start with the gradient of the NPP loss function $L_{NPP}$ i.e. the negative log-likelihood, defined in equation 2

$$\frac{\partial}{\partial \xi} L_{NPP} = \frac{\partial}{\partial x_{Q_i}} \cdot \frac{\partial x_{Q_i}}{\partial \xi} L_{NPP} = \frac{\partial x_{Q_i}}{\partial \xi} \left( -\sum_{i=1}^{n} \frac{\partial}{\partial x_{Q_i}} \log \left( P_{\xi}^{(X_{\mathbf{Q}}, C)}(x_{Q_i}) \right) \right)$$

$$= \frac{\partial x_{Q_i}}{\partial \xi} \left( -\sum_{i=1}^{n} \frac{1}{\left( P_{\xi}^{(X_{\mathbf{Q}}, C)}(x_{Q_i}) \right)} \frac{\partial}{\partial x_{Q_i}} \left( P_{\xi}^{(X_{\mathbf{Q}}, C)}(x_{Q_i}) \right) \right)$$

Here, we remark that $\frac{\partial x_{Q_i}}{\partial \xi}$ will be carried out by back-propagation and the expression after it is the initial gradient.

Next, we derive the gradient of the logical entailment loss function $L_{ENT}$, as defined in equation 7. One estimates the gradient as follows

$$\frac{1}{n} \frac{\partial}{\partial p} L_{ENT} \geq -\frac{1}{n} \sum_{i=1}^{n} \frac{\partial \log(P_{\Pi(\theta)}(Q_i))}{\partial p} \cdot \log(P^{(X_{\mathbf{Q}}, C)}(x_{Q_i})),$$

whereby

- $X_{\mathbf{Q}}$ is the set of random variables associated with the set of the queries $\mathbf{Q}$,
- $x_{Q_i}$ is a training sample, a realization of the set of random variables $X_{\mathbf{Q}}$ associated with the particular query $Q_i$,
- $P^{(X_{\mathbf{Q}}, C)}(x_{Q_i})$ is the probability of the realization $x_{Q_i}$ estimated by the NPP modelling the joint over the set $X_{\mathbf{Q}}$ and $C$ – the set of classes (the domain of the NPP),
- $\log(P_{\Pi(\theta)}(Q_i))$ – the probability of the query $Q_i$ under the program $\Pi(\theta)$ calculated by SLASH (for the reference see the equation (5)),
- and $\frac{\partial \log(P_{\Pi(\theta)}(Q_i))}{\partial p}$ is the gradient as defined in Eq.9.

We begin with the definition of the **cross-entropy** for two vectors $y_i$ and $\hat{y}_i$:

$$H(y_i, \hat{y}_i) := \sum_{j=1}^{m} y_{ij} \cdot \log \left( \frac{1}{\hat{y}_{ij}} \right) = \sum_{j=1}^{m} \left( y_{ij} \cdot \underbrace{\log(1)}_{=0} - y_{ij} \cdot \log(\hat{y}_{ij}) \right) = -\sum_{j=1}^{m} y_{ij} \cdot \log(\hat{y}_{ij}).$$

Hereafter we substitute

$$y_i = \log(P_{\Pi(\theta)}(Q_i)) \qquad \text{and} \qquad \hat{y}_i = P^{(X_{\mathbf{Q}}, C)}(x_{Q_i})$$

and obtain

$$H(y_i, \hat{y}_i) = H \left( \log(P_{\Pi(\theta)}(Q_i)), P^{(X_{\mathbf{Q}}, C)}(x_{Q_i}) \right) = -\sum_{j=1}^{m} \log(P_{\Pi(\theta)}(Q_{ij})) \cdot \log \left( P^{(X_{\mathbf{Q}}, C)}(x_{Q_{ij}}) \right).$$
(11)

We remark that $m$ represent the number of classes defined in the domain of an NPP. Now, we differentiate the equation (11) with the respect to $p$ depicted as in Eq. 9 to be the label of the probability of an atom $c = v$ in $r^{npp}$, denoting $P_{\Pi(\theta)}(c = v)$. Since differentiation is linear, the product rule is applicable directly:

$$\frac{\partial}{\partial p} H(y_i, \hat{y}_i) = -\sum_{j=1}^{m} \left[ \frac{\partial \log(P_{\Pi(\theta)}(Q_{ij}))}{\partial p} \cdot \log \left( P^{(X_{\mathbf{Q}}, C)}(x_{Q_{ij}}) \right) \right.$$

$$\left. + \log(P_{\Pi(\theta)}(Q_{ij})) \cdot \frac{\partial \log \left( P^{(X_{\mathbf{Q}}, C)}(x_{Q_{ij}}) \right)}{\partial p} \right].$$

We do not wish to consider the latter term of $\log(P_{\Pi(\theta)}(Q_i)) \cdot \frac{\partial \log\left(P^{(X_{\mathbf{Q}},C)}(x_{Q_i})\right)}{\partial p}$ because it represents the rescaling and to keep the first since SLASH procure $\frac{\partial \log(P_{\Pi(\theta)}(Q_i))}{\partial p}$ following Eq. 9. To achieve this, we estimate equation from above downwards as

$$\frac{\partial}{\partial p} H\left(y_i, \hat{y}_i\right) \geq -\sum_{j=1}^{m} \frac{\partial \log(P_{\Pi(\theta)}(Q_{ij}))}{\partial p} \cdot \log\left(P^{(X_{\mathbf{Q}},C)}(x_{Q_{ij}})\right). \tag{12}$$

Furthermore, let us recall that under i.i.d assumption we obtain from the definition of likelihood

$$LH(y, \hat{y}) = \prod_{i=1}^{n} LH(y_i, \hat{y}_i),$$

and following the negative likelihood coupled with the knowledge that the log-likelihood of $y_i$ is the log of a particular entry of $\hat{y}_i$

$$L_{ENT} = -\log LH(y, \hat{y}) = -\sum_{i=1}^{n} \log LH(y_i, \hat{y}_i) = -\sum_{i=1}^{n}\sum_{j=1}^{m} y_{ij} \cdot \log(\hat{y}_{ij}) =$$

$$\sum_{i=1}^{n}\left[-\sum_{j=1}^{m} y_{ij} \cdot \log(\hat{y}_{ij})\right] = \sum_{i=1}^{n} H(y_i, \hat{y}_i).$$

Finally, we obtain the following estimate applying inequality (12)

$$\frac{1}{n}\frac{\partial}{\partial p}L_{ENT} = \frac{1}{n}\sum_{i=1}^{n}\frac{\partial}{\partial p}H(y_i, \hat{y}_i) \geq -\frac{1}{n}\sum_{i=1}^{n}\frac{\partial \log(P_{\Pi(\theta)}(Q_i))}{\partial p} \cdot \log\left(P^{(X_{\mathbf{Q}},C)}(x_{Q_i})\right)$$

Also, we note that the mathematical transformations listed above hold for any type of NPP and the task dependent queries (NN with Softmax, PC or PC jointly with NN). The only difference will be the second term, i.e., $\log(P^{(C|X_{\mathbf{Q}})}(x_{Q_{ij}}))$ or $\log(P^{(X_{\mathbf{Q}}|C)}(x_{Q_{ij}}))$ depending on the NPP and task. The NPP in a form of a single PC modeling the joint over $X_{\mathbf{Q}}$ and $C$ was depicted to be the example. With that, the derivation of gradients for both loss functions 2 and 7 is complete, and the training is carried out by coordinated descent.

**Backpropagation for joint NN and PC NPPs:** If within the SLASH program, $\Pi(\boldsymbol{\theta})$, the NPP forwards the data tensor through a NN first, i.e., the NPP models a joint over the NN's output variables by a PC, then we rewrite (8) to

$$\sum_{i=1}^{n}\frac{\partial \log\left(P_{\Pi(\boldsymbol{\theta})}(Q_i)\right)}{\partial \boldsymbol{\theta}} = \sum_{i=1}^{n}\frac{\partial \log\left(P_{\Pi(\boldsymbol{\theta})}(Q_i)\right)}{\partial \mathbf{p}} \times \frac{\partial \mathbf{p}}{\partial \boldsymbol{\theta}} \times \frac{\partial \boldsymbol{\theta}}{\partial \boldsymbol{\gamma}}. \tag{13}$$

Thereby, $\boldsymbol{\gamma}$ is the set of the NN's parameters and $\frac{\partial \boldsymbol{\theta}}{\partial \boldsymbol{\gamma}}$ is computed by the backward propagation through the NN.

## B   APPENDIX B – SLASH PROGRAMS

Here, the interested reader will find the SLASH programs which we compiled for our experiments.
Figure 4 presents the one for the MNIST Addition task, Figure 6 – for the set prediction task with
slot attention encoder and the subsequent CoGenT test. Note the use of the "+" and "-" notation for
indicating whether a random variable is given or being queried for.

```
1  # Define images
2  img(i1). img(i2).
3  # Define Neural-Probabilistic Predicate
4  npp(digit(X), [0,1,2,3,4,5,6,7,8,9]) :- img(X).
5  # Define the addition of digits given two images and the resulting sum
6  addition(A, B, N) :- digit(+A, -N1), digit(+B, -N2), N = N1 + N2.
```

Figure 4: SLASH Program for MNIST addition. The same program was used for the training with
missing data.

```
1  # Is 7 the sum of the digits in img1 and img2?
2  :- addition(image_id1, image_id2, 7)
```

Figure 5: Example SLASH Query for MNIST addition. The same type of query was used for the
training with missing data

```
1   # Define slots
2   slot(s1). slot(s2). slot(s3). slot(s4).
3   # Define identifiers for the objects in the image
4   # (there are up to four objects in one image).
5   obj(o1). obj(o2). obj(o3). obj(o4).
6   # Assign each slot to an object identifier
7   {assign_one_slot_to_one_object(X, O): slot(X)}=1 :- obj(O).
8   # Make sure the matching is one-to-one between slots
9   # and objects identifiers.
10  :- assign_one_slot_to_one_object(X1, O1),
11     assign_one_slot_to_one_object(X2, O2),
12     X1==X2, O1!=O2.
13  # Define all Neural-Probabilistic Predicates
14  npp(color_attr(X), [red, blue, green, grey, brown,
15             magenta, cyan, yellow, bg]) :- slot(X).
16  npp(shape_attr(X), [circle, triangle, square, bg]) :- slot(X).
17  npp(shade_attr(X), [bright, dark, bg]) :- slot(X).
18  npp(size_attr(X), [big,small,bg]) :- slot(X).
19  # Object O has the attributes C and S and H and Z if ...
20  has_attributes(O, C, S, H, Z) :- slot(X), obj(O),
21                  assign_one_slot_to_one_object(X, O),
22                  color(+X, -C), shape(+X, -S),
23                  shade(+X, -H), size(+X, -Z).
```

Figure 6: SLASH Program for ShapeWorld4. The same program was used for the CoGenT experiments.

```
1  # Does object o1 have the attributes red, circle, bright, small?
2  :- has_attributes(o1, red, circle, bright, small)
```

Figure 7: Example SLASH Query for ShapeWorld4 experiments. In other words, this query corresponds to asking SLASH: "Is object 1 a small, bright red circle?".

## C APPENDIX C – EXPERIMENTAL DETAILS

### C.1 SHAPEWORLD4 GENERATION

The ShapeWorld4 and ShapeWorld4 CoGenT data sets were generated using the original scripts of (Kuhnle & Copestake, 2017) (`https://github.com/AlexKuhnle/ShapeWorld`). The exact scripts will be added together with the SLASH source code.

### C.2 AVERAGE PRECISION COMPUTATION (SHAPEWORLD4)

For the baseline slot encoder experiments on ShapeWorld4 we measured the average precision score as in Locatello et al. (2020). In comparison to the baseline slot encoder, when applying SLASH Attention, however, we handled the case of a slot not containing an object, e.g. only background variables, differently. Whereas Locatello et al. (2020) add an additional binary identifier to the multi-label ground truth vectors, we have added a background (bg) attribute to each category (*cf.* Fig. 6). A slot is thus considered to be empty (i.e. not containing an object) if each NPP returns the maximal conditional probability for the *bg* attribute.
As the ShapeWorld4 prediction task only included discrete object properties both for Slot Attention as well as for SLASH Attention the distance threshold for the average precision computation was infinity (thus corresponding to no threshold).

### C.3 MODEL DETAILS

For those experiments using NPPs with PC we have used Einsum Networks (EiNets) for implementing the probabilistic circuits. EiNets are a novel implementation design for SPNs introduced by Peharz et al. (2020) that minimize the issue of computational costs that initial SPNs had suffered. This is accomplished by combining several arithmetic operations via a single monolithic einsum-operation.

For all experiments, the ADAM optimizer (Kingma & Ba, 2015) with $\beta1 = 0.9$ and $\beta2 = 0.999$, $\epsilon = 1e-8$ and no weight decay was used.

**MNIST-Addition Experiments** For the MNIST-Addition experiments, we ran the DeepProbLog and NeurASP programs with their original configurations, as stated in (Manhaeve et al., 2018) and (Yang et al., 2020), respectively. For the SLASH MNIST-Addition experiments, we have used the same neural module as in DeepProbLog and NeurASP, when training SLASH with the neural NPP (SLASH (DNN)) represented in Tab. 2. When using a PC NPP (SLASH (PC)) we have used an EiNet with the Poon-Domingos (PD) structure (Poon & Domingos, 2011) and normal distribution for the leafs. The formal hyperparameters for the EiNet are depicted in Tab. 3.

The learning rate and batch size for the DNN were 0.005 and 100, for DeepProbLog, NeurASP and SLASH (DNN). For the EiNet, these were 0.01 and 100.

Table 2: Neural module – LeNet5 for MNIST-Addition experiments.

| Type | Size/Channels | Activation | Comment |
|---|---|---|---|
| Encoder | - | - | - |
| Conv 5 x 5 | 1x28x28 | - | stride 1 |
| MaxPool2d | 6x24x24 | ReLU | kernel size 2, stride 2 |
| Conv 5 x 5 | 6x12x12 | - | stride 1 |
| MaxPool2d | 16x8x8 | ReLU | kernel size 2, stride 2 |
| Classifier | - | - | - |
| MLP | 16x4x4,120 | ReLU | - |
| MLP | 120,84 | ReLU | - |
| MLP | 84,10 | - | Softmax |

**ShapeWorld4 Experiments** For the baseline slot attention experiments with the ShapeWorld4 data set we have used the architecture presented in Tab. 4. For further details on this, we refer to the original work of Locatello et al. (2020). The slot encoder had a number of 4 slots and 3 attention iterations over all experiments.

Table 3: Probabilistic Circuit module – EiNet for MNIST-Addition experiments.

| Variables | Width | Height | Number of Pieces | Class count |
|-----------|-------|--------|------------------|-------------|
| 784 | 28 | 28 | [4,7,28] | 10 |

Table 4: Baseline slot encoder for ShapeWorld4 experiments.

| Type | Size/Channels | Activation | Comment |
|------|---------------|------------|---------|
| Conv 5 x 5 | 32 | ReLU | stride 1 |
| Conv 5 x 5 | 32 | ReLU | stride 1 |
| Conv 5 x 5 | 32 | ReLU | stride 1 |
| Conv 5 x 5 | 32 | ReLU | stride 1 |
| Position Embedding | - | - | - |
| Flatten | axis: [0, 1, 2 x 3] | - | flatten x, y pos. |
| Layer Norm | - | - | - |
| MLP (per location) | 32 | ReLU | - |
| MLP (per location) | 32 | - | - |
| Slot Attention Module | 32 | ReLU | - |
| MLP | 32 | ReLU | - |
| MLP | 16 | Sigmoid | - |

For the SLASH Attention experiments with ShapeWorld4 we have used the same slot encoder as in Tab. 4, however, we replaced the final MLPs with 4 individual EiNets with Poon-Domingos structure (Poon & Domingos, 2011). Their hyperparameters are represented in Tab. 5.

Table 5: Probabilistic Circuit module – EiNet for ShapeWorld4 experiments.

| EiNet | Variables | Width | Height | Number of Pieces | Class count |
|-------|-----------|-------|--------|------------------|-------------|
| Color | 32 | 8 | 4 | [4] | 9 |
| Shape | 32 | 8 | 4 | [4] | 4 |
| Shade | 32 | 8 | 4 | [4] | 3 |
| Size | 32 | 8 | 4 | [4] | 3 |

The learning rate and batch size for SLASH Attention were 0.01 and 512, for ShapeWorld4 and ShapeWorld4 CoGenT. The learning rate for the baseline slot encoder were 0.0004 and 512.

# D   APPENDIX D - ADDITIONAL RESULTS ON MNIST ADDITION

Table 6: Additional MNIST Addition Results. Test accuracy corresponds to the percentage of correctly classified test images. Both models (DeepProbLog and SLASH (PC)) were trained on the full MNIST data, but tested on images with missing pixels. Test accuracies are presented in percent. The amount of missing data was varied between 50% and 97% of the pixels per image.

|     | **DeepProbLog** | **SLASH** (PC) |
|-----|-----------------|----------------|
| 50% | $\mathbf{79.94 \pm 7.2}$ | $72.2 \pm 12.15$ |
| 80% | $31.6 \pm 6.08$ | $\mathbf{44.2 \pm 8.23}$ |
| 90% | $16.94 \pm 1.76$ | $\mathbf{29.6 \pm 5.77}$ |
| 97% | $12.33 \pm 0.47$ | $\mathbf{17.6 \pm 2.97}$ |

In addition to the setting considered in Evaluation 2, we adjusted the settings for the MNIST Addition task with missing data into the following way.

Training was performed with the full MNIST data set, however the test data set contained different rates of missing pixels. Whereas using an NPP with a PC allows, among other things, to compute marginalization "out of the box" without requiring an update to the architecture or a retraining, this is not so trivial for purely neural-based predicates as in DeepProbLog. Thus, we allowed SLASH (PC) to marginalize over the missing pixels, where this was not directly possible for DeepProbLog.

The results can be seen in Tab. 6 for $50\%$, $80\%$, $90\%$ and $97\%$ of missing pixels per image in the test set. We observe that at $50\%$, DeepProbLog outperforms SLASH by a small margin. For all other rates, we observe that SLASH (PC) reaches significantly higher test accuracies than DeepProbLog. However, we remark that in this setting, SLASH produces larger standard deviations in comparison to DeepProbLog. These results indicate that the conclusions, drawn in the main part of our work, remain true also in this setting of handling missing data.

