# OpenReview forum: "SLASH: Embracing Probabilistic Circuits into Neural Answer Set Programming"
_ICLR.cc/2022/Conference — ICLR 2022 Submitted_

### Official Review · Reviewer_bMeH · 2021-10-18

**Correctness:** 3
**Technical Novelty And Significance:** 2
**Empirical Novelty And Significance:** 2
**Recommendation:** 3
**Confidence:** 5

**Main Review:**

The idea of using PC instead of NN is novel to the best of my knowledge. It leads to interesting experimental results. However, the contribution seems rather incremental with respect to NeurASP and DeepProbLog and the experimental results leave some questions unanswered.

Basically, the only difference of SLASH with respect to NeurASP and DeepProbLog is the possibility of having neural predicates connect to probabilistic circuits instead of NN. While the practical effects of such a possibility may be interesting, the experimental part in my opinion is not sufficient for assessing them.

The speed-up with respect to DeepProbLog in the MNIST-Addition seems to be due to a non-optimal use of hardware resources: while SLASH is parallelized, DeepProbLog is not, even if the derivation for different examples could be run in parallel there as well. The authors should either parallelize DeepProbLog or remove parallelization in SLASH to fairly compare the two.

In the MNIST-Addition task with missing pixel, the comparison with DeepProbLog where the NN interprets missing pixels as background is not fair, I believe an extra Boolean input per pixel should be added to the NN expressing whether the pixel is observed or not or other similar techniques for handling missing data in NN should be used.

In the ShapeWorld4, the authors should report the precise definition of Precision they used, since in (Locatello et al 2020) the precision depends also on a threshold on the distance between the prediction and a matching object. What was the threshold used for Slot Attention? Moreover, in the appendix the author state that they used a different approach for encoding empty slots, where that a slot is considered empty if all neural predicates assign a high probability to an extra value for all the attributes: what was the threshold used in this case to asses whether a probability is high and how was it chosen?

The sentence "this trend still holds even when subtracting the higher precision scores observed in Condition (A)." is not clear: which predictions were removed and how were they chosen?

The theoretical part on semantics and learning is missing some important aspects and looks imprecise in various places.

First of all the authors introduce Neural-Probabilistic Predicates (NPP) without specifying how probabilities are assigned to the individual c=v predicates. For NN, one guesses from NeurASP and DeepProbLog papers that the output of the final softmax layer is used. For PC, this is less clear, as PCs have a single output that returns the joint probability of the configurations of variables at their input. How do you turn a single real value to a discrete distribution? Do you have multiple outputs sharing part of the circuit below them? This should be clarified.

The statement that the use of PCs allows the user to compute also P(X|C) and P(X,C) besides P(C|X) is not fully spelled out: from the probabilities computed by the PC, the semantics assigns an unconditional probability to queries, how do these different probability estimates influence the probability of queries?

Definition 3 states that the probability of a set of queries is given by the product of the probabilities of the individual queries: this is true only if no pair of queries depends on the same NPP. If the queries are really different examples in a learning problem, the independence assumption may hold but in general it does not.

In the definition of NPP the authors use a single argument for h while in the examples color_attr appears as color_attr(1,X): what is the meaning of 1?

The theoretical treatment of learning is confusing: the authors state that they want to maximize the log likelihood of a set of queries but then they state that they learn from a set of pairs (Query,hat p) where they want to match the probability assigned by the model with the one specified in the pair. The two statements seem to contradict each other. Moreover, the loss (equation 9) has a very strange form: it should provide the penalty incurred for example Q with the given probability hat p but in eq 9 hat p is absent... Eq 9 furthermore contains a gradient whose meaning is unclear: the gradient should appear in the derivatives of the objective function as a function of the parameters. Eq 9 also includes term f_\theta^(C_j,X)(x) that is not explained.

Eq 9 seems more the expression of the overall gradient over the examples than the loss.

Proposition 1 is missing a proof or a reference where the proof can be found.

The bibliography contains errors.
Reference

Jiayuan Mao, Chuang Gan, Pushmeet Kohli, Joshua B. Tenenbaum, and Jiajun Wu. The neurosymbolic
concept learner: Interpreting scenes, words, and sentences from natural supervision. In
7th International Conference on Learning Representations, ICLR 2019, New Orleans, LA, USA,
May 6-9, 2019. OpenReview.net, 2019b.

is repeated.
The reference to NeurASP has a wrong title

Zhun Yang, Adam Ishay, and Joohyung Lee. Einsum networks: Fast and scalable learning of tractable
probabilistic circuits. In IJCAI, 2020.

fol->FOL

**Summary Of The Paper:**

The paper presents an approach for neuro-symbolic integration that differs from NeurASP because NN can be replaced by probabilistic circuits (PC) or by NN+PC.

The resulting system, SLASH, is applied to the problems of MNIST-Addition and set prediction on a variant of the ShapeWorld dataset.

The results on MNIST-Addition show that SLASH+NN is faster and more accurate than DeepProbLog and NeurASP. Moreover, when pixels are missing, SLAHS+PC has much better performance than DeepProbLog.

On ShapeWorld, SLASH is compared with Slot Attention (Locatello et al 2020) and found to have better avg precision especially in handling novel combinations of attributes that were not seen during training.

**Summary Of The Review:**

The paper presents an incremental advance over the SotA that seems to have important practical consequences but the experiments do not clearly show them and the theoretical treatment is incomplete and imprecise.

---

> ### Author Response · Authors · 2021-11-15
> **Answers to the questions raised by reviewer bMeH - Part III**
>
> **10.** Eq $9$ seems more the expression of the overall gradient over the examples than the loss.\
> **Answer:** Though Eq $9$ is correct, in fact, it is the derivation of the loss term $L_{ENT}$. We apologize for the misconception. We have updated the paper accordingly. Please find on pages $5-6$ the updated description of the loss functions, $L_{NPP}$ and $L_{ENT}$, inclusive the total one $L_{SLASH}$ as well as detailed derivations of their gradients in appendix A, pp. $13-14$.
>
> **11.** Proposition $1$ is missing a proof or a reference where the proof can be found.\
> **Answer:** In fact, Proposition 1 should not be written as a proposition but as a definition. We have updated the paper accordingly.
>
> **12.** The bibliography contains errors.\
> **Answer:** Thank you very much for pointing this out. We had already noticed ourselves after submission and had updated accordingly.

---

> > ### Comment · Reviewer_bMeH · 2021-11-17
> > **Answers to author's comments**
> >
> > **11** What you call a definition it should not be such: it gives the derivative of a probabilistic model with respect to a parameter, you cannot define the derivative, you have to compute it.

---

> > > ### Author Response · Authors · 2021-11-17
> > > **Answers to the further questions raised by reviewer bMeH - Part III**
> > >
> > > **11.** What you call a definition it should not be such: it gives the derivative of a probabilistic model with respect to a parameter, you cannot define the derivative, you have to compute it.\
> > > **Answer:** Since the term $\tfrac{\partial\log\left( P_{\Pi(\mathbf{\theta})}(Q)\right)}{\partial\mathbf{p}}$ follows NeurASP, we stated this as a definition. Please see page 6 for the remark. However, in the newest update of the paper we now handle this as an equation (p. 6, Eq. 9 in new version).

---

> ### Author Response · Authors · 2021-11-15
> **Answers to the questions raised by reviewer bMeH - Part II**
>
> **5.** First of all the authors introduce Neural-Probabilistic Predicates (NPP) without specifying how probabilities are assigned to the individual $c=v$ predicates. For NN, one guesses from NeurASP and DeepProbLog papers that the output of the final softmax layer is used. For PC, this is less clear, as PCs have a single output that returns the joint probability of the configurations of variables at their input. How do you turn a single real value to a discrete distribution? Do you have multiple outputs sharing part of the circuit below them? This should be clarified.\
> **Answer:** Maybe Figure 1 is a bit misleading. Please accept our apologies. In the updated version, you will find a new version of Figure 1. We wish to explain it in more detail in the following. Each PC depicted in the Neural-Probabilistic Predicate models a joint probability of $P(z_i, C)$ at its root node and its children outline class conditional distributions $P(z_i|C=c_j)$. A sample $z_i$ is classified by applying Bayes' rule: $P(C|z) = \tfrac{P(z|C)*P(C)}{P(z)} = \tfrac{P(z|C)*P(C)}{\sum_{c_j}P(z|c_j)P(c_j)}$. The class prior $P(C)$ can be fixed to, e.g., uniform, or learned during the training. For the detailed explanations, we refer to Section 3 of "Random Sum-Product Networks: A Simple and Effective Approach to Probabilistic Deep Learning" by Peharz et al. published at UAI 2019. You are right regarding the DNNs. Indeed, we are using the final softmax layer.
>
> **6.** The statement that the use of PCs allows the user to compute also $P(X|C)$ and $P(X,C)$ besides $P(C|X)$ is not fully spelled out: from the probabilities computed by the PC, the semantics assigns an unconditional probability to queries, how do these different probability estimates influence the probability of queries?\
> **Answer:** Please have a look at the answer to the previous question. Further, we refer to definitions 1,2 and 3, as well accompanying examples in "Lecture Notes Probabilistic Circuits: Representation and Inference" by Choi at al, AAAI2020 for additional sources.
>
> **7.** Definition 3 states that the probability of a set of queries is given by the product of the probabilities of the individual queries: this is true only if no pair of queries depends on the same NPP. If the queries are really different examples in a learning problem, the independence assumption may hold but in general it does not.\
> **Answer:** Please note that we make an explicit independence assumption here to keep computation easy. In general, we agree that the independencies do not hold. We will make it very clear that this is an assumption in the camera-ready version.
>
> **8.** In the definition of NPP the authors use a single argument for h while in the examples color_attr appears as color_attr(1,X): what is the meaning of $1$?\
> **Answer:** We understand that this can cause some confusion. The $1$ is an artifact of the ASP solver and can be read as "give me the gradient of the logarithm of the probability of the query $Q$ with the respect to $p$, the label of the probability of an atom $c = v$ in $r^{npp}$, denoting $P_{\Pi(\theta)}(c = v)$". However, as this is syntactic sugar of ASP and less intuitive for the high-level understanding, we will remove this from the example programs of the paper.
>
> **9.** The theoretical treatment of learning is confusing: the authors state that they want to maximize the log likelihood of a set of queries, but then they state that they learn from a set of pairs $(Q, \hat{p})$ where they want to match the probability assigned by the model with the one specified in the pair. The two statements seem to contradict each other. Moreover, the loss (equation $9$) has a very strange form: it should provide the penalty incurred for example Q with the given probability $\hat{p}$ but in eq $9$ $\hat{p}$ is absent... Eq $9$ furthermore contains a gradient whose meaning is unclear: the gradient should appear in the derivatives of the objective function as a function of the parameters. Eq $9$ also includes term $f_{\theta}^{(C_j,X)}(x)$ that is not explained.\
> **Answer:** We apologize for the misunderstanding, but there is not in fact $\hat{p}$ necessary. Following _learning from entailment_ setting, we actually assume the query to be true over all training instances. Further, we remark that although gradient ascent is used to maximize the log likelihood of a set of queries, the overall training is performed via coordinate descent. That is, the necessity of $\hat{p}$ becomes irrelevant. Closing, we point out that the total loss function stated in the equation (10) entails two different parts. And, the loss function stated in (8) is negative log-likelihood written as sum over cross-entropy plus the extra information provided by SLASH, which is being pushed to the gradient. Please find our updated notation also in the new version on pages 5-6 and the detailed derivation of the total loss function in Appendix A on page 13.

---

> > ### Comment · Reviewer_bMeH · 2021-11-17
> > **Answers to authors' comments**
> >
> > **5** In Section 3 of "Random Sum-Product Networks: A Simple and Effective Approach to Probabilistic Deep Learning" the $P(z_i|C=c_j)$ are the roots of the PC, in your case I don't understand the meaning of your root $P(z_i,C)$: it is not a joint, as you would have a different value for each $c_j$ or you should have the class as an input to the circuit.
> >
> > **6** My question is: given that you can compute $P(X|C)$, $P(C|X)$ and $P(X,C)$, how does this influence the probability of the query of the ASP program? How does it change depending on the value that you extract from the PC?
> >
> > **9** This part of the article has now been completely rewritten but still has issues: you define the log likelihood as a two argument function in eq. 3 and 8, but the log likelihood depends only on the probability of the data. The loss is a two argument function, you seem to confuse loss with log likelihood.
> > In eq. 3, you do not say what $P_\xi^{(X,C)}(x_i)$ is.
> > In eq 8 you do not say what  $P^{(X_Q,C)}(x_Q)$ is.

---

> > > ### Author Response · Authors · 2021-11-17
> > > **Answers to the further questions raised by reviewer bMeH - Part II**
> > >
> > > **5.** In Section 3 of "Random Sum-Product Networks: A Simple and Effective Approach to Probabilistic Deep Learning" the $P(z_i|C=c_j)$ are the roots of the PC, in your case I don't understand the meaning of your root $P(z_i,C)$: it is not a joint, as you would have a different value for each $c_j$ or you should have the class as an input to the circuit.\
> > > **Answer:** The overall root node, as defined in Figure 1 can be written as $P(z_i,C) = \sum_{j}P(z_i|C=c_j)*P(C=c_j)$, thereby $P(C=c_j)$ (prior) are weights between the root node $P(z_i,C)$ and its children $P(z_i|C=c_j)$. We write the values of the overall root node's children as $P(z_i|C=c_j)*P(C=c_j) = P(z_i, C=c_j)$ following the factorization theorem. Computing the root node in this way does not harm the validity of PCs. We hope that this might clarify the misunderstanding. We have additionally readjusted Figure 1.
> > >
> > > **6.** My question is: given that you can compute $P(X|C)$, $P(C|X)$ and $P(X,C)$, how does this influence the probability of the query of the ASP program? How does it change depending on the value that you extract from the PC?\
> > > **Answer:** Ok, now we understand the intent of the question better. Thank you for clarifying. As formulated in definition 3 on page 5, the probability of the query $Q$ is nothing else than the sum of the probabilities of all potential solutions $I$ which satisfy the query. Further, definition 2 specifies the probability of a potential solution as the fraction between the product of the probabilities of all atoms $v = c$ in $I| {r^{npp}} $, where ${r^{npp}}$ represents the set of all logic rules concearning NPPs, and the number of possible solutions of the program $\Pi$ agreeing with in $I| {r^{npp}}$. Thus, the computations of $P(X|C)$, $P(C|X)$ and $P(X,C)$ influence the probabilities of all atoms $v = c$ in $I| {r^{npp}}$ in a similar fashion. We hope this clarifies your question.
> > >
> > > **9.** This part of the article has now been completely rewritten but still has issues: you define the log likelihood as a two argument function in eq. 3 and 8, but the log likelihood depends only on the probability of the data. The loss is a two argument function, you seem to confuse loss with log likelihood. In eq. 3, you do not say what $P_{\xi}^{(X,C)}(x_i)$ is. In eq 8 you do not say what $P^{(X_{\mathbf{Q}},C)}(x_{\mathbf{Q}})$ is.\
> > > **Answer:** We here go through the several sub-remarks one by one. Firstly, indeed, Eq 3 (2 in the updated version) expresses the negative log-likelihood, which depends solely on the probability of the data. However, this is not the case for the Eq 8 (7 in the updated version). We would like to emphasize that training overall in SLASH is carried out by coordinate descent. However, for Eq 8 (7), the parameters of the NPP are frozen and the negative log-likelihood of the query $Q_i$ is, in fact, being calculated by ASP. Thus, it is a result of probabilistic logical reasoning.
> > > Secondly, we politely disagree on confusing loss with likelihood. However, we understand that our overloaded meaning of $L$ can lead to confusions regarding which of our terms are loss functions and which are log-likelihood computations. Please accept our apologies for the somewhat misleading notations between the _losses_ $L_{NPP}$, $L_{ENT}$, $L_{SLASH}$ and the _log-likelihood_ function, $L(\cdot,\cdot)$. To prevent further confusions, we have updated the notations for likelihood from $L$ to $LH$. Furthermore, $LH(\cdot,\cdot)$ contains two arguments due to the derivation from the cross entropy presented in the Appendix A, pp 13-14. Thirdly, in Eq 3 (2 in the updated version) $P_{\xi}^{(X,C)}$ are the predictions $\hat{x_i}$ obtained from the PC encoded in the NPP. Please see p 5 in the updated version for the further details. Lastly, In Eq 8 (7 in the updated version) is $P^{(X_{\mathbf{Q}},C)}(x_{\mathbf{Q}})$ the probability of the set of realizations $x_{\mathbf{Q}}$ estimated by the NPP that models the joint probability over the set $X_{\mathbf{Q}}$ and $C$ (the set of classes).

---

> ### Author Response · Authors · 2021-11-15
> **Answers to the questions raised by reviewer bMeH - Part I**
>
>
> We thank you for your time and remarks. Below you can find our answers to the mentioned issues:
>
> **1.** The speed-up with respect to DeepProbLog in the MNIST-Addition seems to be due to a non-optimal use of hardware resources: while SLASH is parallelized, DeepProbLog is not, even if the derivation for different examples could be run in parallel there as well. The authors should either parallelize DeepProbLog or remove parallelization in SLASH to fairly compare the two.\
> **Answer:** We apologize that the reviewer finds the results in Table $1$ unclear. During the training of DeepProbLog, the GPU was used for the DNN. The same goes for NeurASP, SLASH (DNN) and SLASH (PC).  I.e., the same hardware resources were used for all three DPPLs in this regard: GPU for low-level perception (deep learning) and CPU for high-level logical reasoning. The difference between the three DPPLs is that SLASH treats queries batchwise, compared to DeepProbLog and NeurASP. This allows to parallelize the ASP calls and so gives the rise to the speed-up described in Table $1$ and is used in SLASH Attention for the set prediction task. We will make this clear in the paper.
>
> **2.** In the MNIST-Addition task with missing pixel, the comparison with DeepProbLog where the NN interprets missing pixels as background is not fair, I believe an extra Boolean input per pixel should be added to the NN expressing whether the pixel is observed or not or other similar techniques for handling missing data in NN should be used.\
> **Answer:** Thanks for this idea, but there seems to be a misunderstanding here. SLASH introduces probabilistic circuits (PCs) into deep programming languages. To illustrate that this is a sensible idea, we use them to model the distribution over the pixels and the class prediction. This way, we can easily marginalize missing pixels out. Indeed, DeepProbLog cannot easily do this when using NNs, since a standard NN is not encoding a joint distribution. We could have used an autoregressive NN encoding a joint distribution instead, but then we are loosing tractability. If we add tractability, we essentially arrive at our experimental setup.\
> The point of the experiment was to show that a plain DNN will treat that missing pixels as background, whereas a PC can handle these as special cases (i.e., missing data) per design. In other words, the intention was thus explicitly to show that a purely neural network requires techniques such as the one you have mentioned, whereas with the use of PCs this comes somewhat "for free".
> Following your suggestion, we rerun MNIST Addition task with $97\%$ drop-out rate annotating with $-1$ the missing pixels within DeepProbLog.
> Using five different seeds ($1-5$), the achieved test accuracy by the end of the training was $16.95 \pm 4.45$.
>
> **3.** In the ShapeWorld4, the authors should report the precise definition of Precision they used, since in (Locatello et al. 2020) the precision depends also on a threshold on the distance between the prediction and a matching object. What was the threshold used for Slot Attention? Moreover, in the appendix the author state that they used a different approach for encoding empty slots, where that a slot is considered empty if all neural predicates assign a high probability to an extra value for all the attributes: what was the threshold used in this case to asses whether a probability is high and how was it chosen?\
> **Answer:** Yes, thank you for pointing this out. We have added this information in our updated version of the paper. Please see the last subsection of the Appendix C.2. As we were not considering the prediction of object locations, but only discrete object properties, the threshold we had used corresponds to the infinite threshold of Locatello et al.
> Concerning the average precision computation for background slots in SLASH Attention, in fact, "high conditional probability" denotes the maximum probability over all attributes of a category. Thus, this computation is handled the same as for baseline Slot Attention. Additionally, the distance threshold for SLASH Attention was also the same as for the baseline Slot Attention, namely infinity.
>
> **4.** The sentence "this trend still holds even when subtracting the higher precision scores observed in Condition (A)." is not clear: which predictions were removed and how were they chosen?\
> **Answer:** We believe there is a misunderstanding here. The essence was to make it clear that the performance advantage that we observe already in Test Condition A (and also that observed in evaluation 3), i.e., an overall improved prediction performance, will most likely also be present in the improved results for Test Condition B. However, even when taking this effect into account we further observe an improved composition generalization performance with SLASH.

---

> > ### Comment · Reviewer_bMeH · 2021-11-17
> > **Answers to the authors' comments**
> >
> > 1. The difference between the three DPPLs is that SLASH treats queries batchwise, compared to DeepProbLog and NeurASP.
> > **Answer**: I still can't see why DeepProbLog cannot be parallelized: even if it treats queries one by one, why can't individual queries be run in parallel?

---

> > > ### Author Response · Authors · 2021-11-17
> > > **Answers to the further questions raised by reviewer bMeH - Part I**
> > >
> > > We thank the reviewer for the discussions and answer the questions pointwise.
> > >
> > > **1.** I still can't see why DeepProbLog cannot be parallelized: even if it treats queries one by one, why can't individual queries be run in parallel?\
> > > **Answer:** Yes, we fully agree that DeepProbLog can be run in parallel, which would lead to a decreased RTE than we have specified. Although, wish to remark that we believe this not to be an easy task. However, the point we were trying to make with the RTE result of Tab.1a  was that although we are performing tractable probabilistic inference which comes with the many advantages, such as marginalization, joint probabilistic modelling etc. SLASH with PCs does not, in fact, have a disadvantage in computation time. This is of particular relevance as probabilistic models have a reputation of bearing high computation times. This achievement is ultimately a result of the optimal implementation of SLASH (via its batch-wise loss and parallelization). We do agree that this is more of a practical/ engineering feat, than an argument for or against a particular DPPL. However, we consider this a very valuable remark for future applications and research. We see that the way we have presented these results can be misleading and apologize, as this was not our intention, and have updated the paper by removing the RTE values from Tab 1a on page 7.

---

> ### Author Response · Authors · 2021-11-28
> **We have answered your remaining queries !!**
>
> Hello,
>
> We would like to thank the reviewer for engaging in a discussion on 17th November and we had given answers for the remaining concerns on the very same day. We also uploaded the newest version of the paper which we believe alleviates the main concerns of the reviewer (see general comment for changes). Since there is only a 1 day left in the discussion It would be helpful if the reviewer can clarify that the issues were indeed resolved and if yes we would appreciate if the reviewer can change their overall assessment of our work.
>
> Regards,
>
> The Authors

---

### Official Review · Reviewer_TutZ · 2021-10-27

**Correctness:** 3
**Technical Novelty And Significance:** 3
**Empirical Novelty And Significance:** 2
**Recommendation:** 5
**Confidence:** 4

**Main Review:**

Unifying subsymbolic/perceptual and symbolic/logical reasoning is a very active field of research and I found the core idea of this paper
interesting.

The paper is generally well written, but I wish that it focused less on the benefits of neuro-symbolic approaches, which are well known in literature, and more on the semantics of SLASH and the actual benefits of embracing PCs into the framework. I found the paper too hasty on the latter aspects and I think that some examples could definitely help.

The preliminary results reported are promising, although I found them overly enthusiastic at times. In Evaluation 1, the accuracy margin between SLASH and DeepProbLog/NeurASP is way too small to claim the superiority of the method. The positive results on RTE are only due to a parallel implementation rather than on the core idea beyond SLASH and I don't see why NeurASP couldn't implement a similar
technique. The ability of SLASH of handling missing data in a principled way is indeed a welcome feature. I don't see why not reporting
the performance of NeurASP in Evaluation 2 though.

Evaluation 3 and 4 shows that adding a logical component on top of a Slot Attention model is beneficial in the set prediction task. I think that DeepProbLog and NeurASP could also be applied in this setting, by definining the neural predicates on top of the latent representation provided by SA. If that's not the case, it would be useful to mention why.

Minors:

1) "In contrast to query evaluation in Prolog (Colmerauer & Roussel,1996; Clocksin & Mellish, 2003) which may lead to an infinite loop,
many modern answer set solvers use Conflict-Driven-Clause-Learning (CDPL) which, in principle, always terminates."

This is a slightly reworded version of a sentence from the first paragraph of the ASP page on Wikipedia. DPLL (and modern variants such as CDCL) do terminate in principle *and* in practice. Guaranteed termination implies that, in contrast with Prolog, ASP is not Turing complete.

2) "due to SLASH’s semantics, our interest lies in rewarding the right solutions (v = c) and penalizing wrong ones (v != c)"

Isn't this true in most machine learning settings?

3) The function f in the loss term (Eq.9) is not defined.

Typos:

page 2
- end2end -> end-to-end
- IN this way -> In ths way
- CDPL -> CLCL

page 3
- MMC -> WMC

page 5
- referred to as stable model, I, in ASP -> referred to as stable model in ASP

**Summary Of The Paper:**

This paper proposes a neuro-symbolic model, dubbed SLASH, that extends previous deep probabilistic programming languages with probabilistic circuits (PCs). The advantage of this approach with respect to similar frameworks, such as DeepProbLog or NeurASP, lies in defining predicates using Sum-Product Networks (SPNs), a subclass of PCs that admit tractable marginal/conditional inference over any subset of variables. Additionally, neural networks and SPNs can be unified in the same framework, for instance by defining PCs on the latent representation extracted by an Attention Slot module. Experiments on different tasks show promising results for this approach.


**Summary Of The Review:**

- Unifying PCs with neuro-symbolic approaches is an interesting research direction
- The technical description of SLASH could be improved, possibly by shortening the introduction and motivation for neuro-symbolic approaches
- The preliminary results are promising although the empirical evaluation could be improved. Most importantly, the experiments and the following discussion seem to oversell the SLASH framework rather than rigorously investigating the use of PCs in neuro-symbolic models.

---

> ### Author Response · Authors · 2021-11-15
> **Answers to the questions raised by reviewer TutZ - Part II**
>
> **6.** "In contrast to query evaluation in Prolog (Colmerauer \& Roussel,1996; Clocksin \& Mellish, 2003) which may lead to an infinite loop, many modern answer set solvers use Conflict-Driven-Clause-Learning (CDPL) which, in principle, always terminates." This is a slightly reworded version of a sentence from the first paragraph of the ASP page on Wikipedia. DPLL (and modern variants such as CDCL) do terminate in principle and in practice. Guaranteed termination implies that, in contrast with Prolog, ASP is not Turing complete.\
> **Answer:** We would like to point out that further developments of Davis-Putman-Logemann-Loveland procedure are the centerpiece of ASP solvers and those indeed terminate in theory compared to Prolog's SLDNF resolution. Please have a look at the introduction of "What is Answer Set Programming?" by Vladimir Lifschitz, AAAI 2008.
> Further, we wish to remark that, although Prolog is Turing-complete, it is syntactically limited and lacks of reasoning power. In contrast, Answer Set Programming combines the concepts of both logic and constraint programming. For further arguments, we cite the work of Chiaki Sakama and Katsumi Inoue "Answer Set Programming", Journal of Japanese Society for Artificial Intelligence 25(3), pp. 368-378, 2010.
>
> **7.** "due to SLASH’s semantics, our interest lies in rewarding the right solutions (v = c) and penalizing wrong ones (v != c)"
> Isn't this true in most machine learning settings?\
> **Answer:** We respectfully disagree, since there are different settings in machine learning which do not follow the said settings for, e.g., clustering and reinforcement learning.
>
> **8.** The function $f$ in the loss term (Eq.9) is not defined.\
> **Answer:** We apologize for the somewhat misleading notation. In fact, the function $f$ in the loss term (Eq. 9) stands for the density function of the joint over $C$ and X being modelled by the NPP. This is the usual way, how the PCs being implemented and trained. Of course, one can write $P$ instead of an $f$ in the loss term.
>
> **Further remarks from reviewer**
> * The technical description of SLASH could be improved, possibly by shortening the introduction and motivation for neuro-symbolic approaches
> * The preliminary results are promising although the empirical evaluation could be improved. Most importantly, the experiments and the following discussion seem to oversell the SLASH framework rather than rigorously investigating the use of PCs in neuro-symbolic models.
>
> **Answer:** Thank you for your insights, but we respectfully disagree, since we dedicate experiments designed for investigation of the PC's utilization in neuro-symbolic computations. SLASH's purpose is to offer the unifying framework which offers intentionally the simple access to the neuro-symbolic computations. As we have shown, the advantages of the PCs in handling incomplete data (Table 2) combined with the scalability allow us to perform the Set Prediction Task within SLASH. In addition to that, we have successfully shown that SLASH Attention not only performs better regarding mean precision, but is also more stable under Compositional Generalization Test (CoGenT).

---

> ### Author Response · Authors · 2021-11-15
> **Answers to the questions raised by reviewer TutZ - Part I**
>
> We thank you for your time and remarks. Below you can find our answers to the mentioned issues:
>
> **1.** The paper is generally well written, but I wish that it focused less on the benefits of neuro-symbolic approaches, which are well known in literature, and more on the semantics of SLASH and the actual benefits of embracing PCs into the framework. I found the paper too hasty on the latter aspects and I think that some examples could definitely help.\
> **Answer:** We are happy to hear our work to be well written. While the benefits of Neuro-Symbolic approaches are becoming more well known recently (but are not as apparent in all parts of the AI community), we emphasize the ability of SLASH to combine neural, logical and truly tractable probabilistic modules within a unifying framework and via a single optimization procedure. We believe we have presented the benefits of PCs in Probabilistic Logic Programming over several claims. Firstly, via NPPs with PCs we have a much larger set of probabilistic queries, such as $P(X, C)$ and $P(C|X)$ in addition to $P(X|C)$, see pp. 4 - 5. Particularly, we emphasize that neural predicates as used in previous works can in principle only handle queries of the form $P(X|C)$, p. 4. With our results on computation time, we wished to show that this advantage of tractable probabilistic inference comes at the same or fewer costs as previous DPPLs, p. 7. Next, with our experiments on missing data, we explicitly show the practical advantages of PCs, pp. 7-8. Specifically, simply by using the marginalization abilities of PCs, we can handle the case of missing data out of the box. Lastly, with PC we are getting a generative model, which comes in handy in case of small data sets and/or image generation.
>
> **2.** In Evaluation 1, the accuracy margin between SLASH and DeepProbLog/NeurASP is way too small to claim the superiority of the method.\
> **Answer:** Evaluation 1 is the difference of our loss computation in overall prediction performance (seen when comparing the SLASH(DNN) against DeepProbLog and NeurASP) even though using the same DNN architecture as NeurASP and DeepProbLog. Secondly, with evaluation 1 we wanted to show that although we are making use of PCs which perform tractable probabilistic inference (and that come with a much larger set of modelling possibilities) we have no disadvantage in computational costs. In fact, the experimental advantages of PCs are better shown in evaluation 2. We take reviewer's concern and reworded it accordingly. Lastly, we wish to emphasize as in the answer to the previous question that with a PC, we are receiving a generative model which offers interesting opportunities for the future work.
>
> **3.** The positive results on RTE are only due to a parallel implementation rather than on the core idea beyond SLASH and I don't see why NeurASP couldn't implement a similar technique.\
> **Answer:** We wish to emphasize that the main difference between the two DPPLs is that SLASH uses a different loss function and treats queries batchwise, compared to NeurASP and DeepProbLog. This allows for the ASP to be called in parallel and so gives the rise to the speed-up described in Table 1.
>
> **4.** The ability of SLASH of handling missing data in a principled way is indeed a welcome feature. I don't see why not reporting the performance of NeurASP in Evaluation 2, though.\
> **Answer:** NeurASP again uses a neural network predicate, we used in DeepProbLog, as a representative of neural LP. Thus, we do not expect the performance of NeurASP to be better.
>
> **5.** Evaluation 3 and 4 shows that adding a logical component on top of a Slot Attention model is beneficial in the set prediction task. I think that DeepProbLog and NeurASP could also be applied in this setting, by defining the neural predicates on top of the latent representation provided by SA. If that's not the case, it would be useful to mention why.\
> **Answer:** We understand the reviewer's comment but would like to politely disagree here. As shown in Table 1, thanks to the different loss function and the batchwise treatment of queries, compared with DeepProbLog and with NeurASP, SLASH is scalable, which is crucial for evaluations 3 and 4 to be performed in reasonable time compared to the stand-alone training.

---

> ### Author Response · Authors · 2021-11-28
> **It will be helpful if the reviewer can engage in a discussion!!**
>
> Hello,
>
> Thank you for the original review. We had already answered and we believe resolved all your concerns on Nov 15th but there was no response. Now there is only 1 day left for discussion and we believe that it will be helpful if you could engage with us in a discussion since we believe that we have answered all your original queries.
>
> Regards,
>
> The Authors

---

> > ### Comment · Reviewer_TutZ · 2021-11-28
> > **Re**
> >
> > I apologise for the late reply, I'm on the fence with this submission.
> > The introduction of PCs into the neuro-symbolic framework is the main contribution, but its benefits are only showcased in Evaluation 2 (MNIST addition w/ missing data). In my opinion, Evaluation 1 should be titled "SLASH performs comparably to SOTA DPPLs in MNIST addition". The improved scalability with respect to e.g. DeepProblog doesn't derive from a profound scientific insight but from a parallel implementation that in principle other approaches could adopt (unless proven otherwise). Evaluations 3 and 4 only show that integrating neural and symbolic paradigms is beneficial in complex reasoning tasks. I agree with reviewer bMeH that the semantics could be more clearly presented. While in principle the PCs enables flexible generative modeling, it is not clear to me how this is supported effectively by the learning-by-entailment paradigm borrowed from (Deep)ProbLog, i.e. what happens when the queries are defined on different sets of variables.
> > While I think that the core idea behind Slash is interesting and the preliminary results are promising, I think that both the technical part and the experimental section could be improved. This work has the potential of being very impactful, but in the current state it seems rather incremental. I am keeping the same overall score for this submission.

---

> > > ### Author Response · Authors · 2021-11-28
> > > **Answers to the further questions raised by reviewer TutZ**
> > >
> > > Thanks for your response. We would like to point that we rewrote Section 3 concerning SLASH's learning procedure to present semantics more clearly. Additionally, we removed our arguments on scalability from Evaluation 1 (Table 1a on p. 7), since it is not our intention to mislead our readers. Further, we extended Evaluation 2 (MNIST addition w/ missing data) with the further setting. In Appendix D, p. 18, you will find the results for the “opposite” experiment. Both models (DeepProbLog and SLASH (PC)) were trained on the full MNIST data, but tested on images with missing data. The results indicate that the conclusions, drawn in Evaluation 2, remain true also in this setting of handling missing data.\
> > > Further, we would like to ask you to elaborate your seconds question. As it is now, we assume that you are asking what would happen if, for example, two disjoint sets of random variables were considered within a SLASH program. This question can be answered as follows. For each of the sets, one will define two different NPPs. Following MNIST addition, one can convert the task to two data sets: MNIST and SVHN. Such task can be called Number Addition. Because the probabilistic logical conclusions will not change, including the set of possible solutions, the paradigm will also yield the desired results. Furthermore, the generative nature of PCs allows working with small data sets within SLASH. That is, one can generate the queries including the data during training, which makes SLASH very flexible.

---

> > > > ### Comment · Reviewer_TutZ · 2021-11-29
> > > > **Re: Answers to the further questions raised by reviewer TutZ**
> > > >
> > > > > Further, we would like to ask you to elaborate your seconds question.
> > > >
> > > > I think that I found the answer in Appendix A.
> > > > Can you confirm me that queries can be defined on multiple (possibly overlapping) sets of variables X_{Q_I}? I.e. that in
> > > >
> > > > > "x_{Q_i} is a training sample, a realization of the set of random variables X_Q associated with the particular query Q_i"
> > > >
> > > > different training examples could realize different subsets of X and C?

---

> > > > > ### Author Response · Authors · 2021-11-29
> > > > > **Response to the reviewer**
> > > > >
> > > > > Yes you are right. The different training examples can indeed realize different subsets of variables (overlapping as well) based on the query.
> > > > >
> > > > > If this answers all the questions of the reviewer satisfactorily then we would appreciate if the reviewer can reconsider the overall evaluation.

---

### Official Review · Reviewer_8iux · 2021-11-02

**Correctness:** 4
**Technical Novelty And Significance:** 3
**Empirical Novelty And Significance:** 3
**Recommendation:** 8
**Confidence:** 2

**Main Review:**

Even as a non-expert of DPPLs, I have found the SLASH language specification in S3 easy to understand.
In S3.1, I believe that the generative nature of SPNs is crucial and this point should be better stressed.
The number of queries to be computed should be made explicit, as well as the possibility of performing this in parallel (and the same holds for the ASP queries). More generally speaking a better characterisation of the inferential complexity of SLASH should be provided.
Regarding the experiments some words should be spent about the computational efficiency (wrt DeepProbLog) when coping with missing data.


**Summary Of The Paper:**

This paper presents a new deep probabilistic programming language based on ASP. The key feature is the use of a single attention encoder paired with a collection of probabilistic circuits (one for each "category"/model solution of the neuro-symbolic task). ASP is finally used to answer a query given the probabilistic predicates returned by the circuits for the features extracted by the encoder and the logical program modelling the particular task under consideration. SOTA accuracies are achieved with advantages related to the generative nature of the considered approach (e.g., when coping with missing data) and good computation times (due to the parallelisation of the ASP tasks).


**Summary Of The Review:**

A new ASP-based deep probabilistic programming language. Using generative probabilistic circuits allow to cope with missing data. This seems to be a novel approach and SOTA results are achieved. I think the paper can be safely accepted.

---

> ### Author Response · Authors · 2021-11-15
> **Answers to the questions raised by reviewer 8iux**
>
> We thank you for your time and remarks. Below you can find our answers to the mentioned issues:
>
> **1.** In S3.1, I believe that the generative nature of SPNs is crucial, and this point should be better stressed.\
> **Answer:** Thank you for the suggestion and we will look closely into the generative power of SPNs + SLASH in a following work.
>
> **2.** The number of queries to be computed should be made explicit, as well as the possibility of performing this in parallel (and the same holds for the ASP queries).\
> **Answer:** Yes, thank you, we will incorporate it.
>
> **3.** More generally speaking, a better characterization of the inferential complexity of SLASH should be provided.\
> **Answer:** Thank you, we note here that SLASH's inference relies on the underlying ASP solver, which in the worst-case scenario is a NP-hard problem, and in the best-case scenario linear due to the underlying property of the PC's inference being linear in the size of the network.
>
> **4.** Regarding the experiments, some words should be spent about the computational efficiency (wrt DeepProbLog) when coping with missing data.\
> **Answer:** We would like to emphasize that PCs have linear time complexity regarding their size for the inference, and the average Runtime Per Epoch (RTE) was 5 minutes and 14 seconds, during DeepProbLog's has reached 19 minutes and 34 seconds on average.

---

### Official Review · Reviewer_qUza · 2021-11-03

**Correctness:** 3
**Technical Novelty And Significance:** 3
**Empirical Novelty And Significance:** 3
**Recommendation:** 6
**Confidence:** 3

**Main Review:**

the topic of this paper, SLASH has a complex structure. It includes :

- a NPP: which is a standard NN decription language
 Reading the paper, it looks like it can describe any NN, but  you also use
- an ASP program, that computes probabilities by ASP eval.
- Finally, the whole? thing is trained using grad ascent

 - the initial probabilities were obtained from?
- ASP programs have multiple models, which have multiple atoms. \sum Models = 1? P(M1 and M2) ?
- Is it correct to use the word query for  a model?

1. What is the point of the LP? It is not very well motivated.

 It would be nice to have examples showing how to write it?

2. ASP; do you use non-strat neg? Agg? Everyting in your programs?

3.  what goes in the NN is the solver output as a number? Ie, you run the solver every time?

User vs training loss: I got lost, what is the point of a loss function that is not used in training? TensorFlow and friends allow to define a second function on tuning data, is that it?

Evaluation
can you give short info on dataset size at the beginning?
slash(pc) and slash(dnn) have the same ASP?
Table ii: why no ASP only? Such good result with low data suggest a very strong prior,
again, you allow users to define a loss funcion, but not to use it? Maybe I am missing the point.

t SLASH’s batch-wise loss -> confirming? Had you discussed batched evaluation before? Isn't this standard in DNNs?


Small questions:
 gradient for the solution does not have to be positive: zero? This should not be in the appendix.
Attention vs slash attention: you define the latter in page 8?
Please refer to the sw you use in the text (eg, which ASP solver?)

**Summary Of The Paper:**

This paper presents slash a neuro symbolic system; the system presents strong results, but on few datasets

**Summary Of The Review:**

I think this is a nice paper, suggesting the logic program provides strong background data for the network and that the gradient based search can  provide good prob  estimates.

The one flaw in this paper is that you only compare with DeepProblog, You drop NeurASP after the first table In general. it is hard for me to fully accept your claims :(

---

> ### Author Response · Authors · 2021-11-15
> **Answer to the questions raised by reviewer qUza - part II**
>
> **12.** Table ii: why no ASP only?\
> **Answer:** We assume you mean NeurASP here? The goal of the experiment, the results of which are summarized in Table 2, is to show that PCs can handle incomplete data without additional techniques compared to a DNN. Apart from that, the application of NeurASP would not offer any further insights. As can be seen in Table 1, the test accuracy on average is subject to the models trained within NeurASP.
>
> **13.** Such good result with low data suggest a very strong prior, again, you allow users to define a loss funcion, but not to use it? Maybe I am missing the point.\
> **Answer:** Having the good prior is the point if this experiment. It is being learned during the training for SLASH (PC). Maybe there is a misunderstanding. Please accept our apologies. We are not allowing users to define a loss function. Our point is the opposite. One loss function to rule them all ;-), i.e., the tasks to be solved.
>
> **14.** SLASH’s batch-wise loss -> confirming?\
> **Answer:** If we understand your question correctly, yes SLASH's loss is batch-wise. SLASH's batch-wise loss paves the way for the parallelization of the ASP-calls and so the scalability necessary for Set Prediction Task being performed in reasonable time.
>
> **15.** Had you discussed batched evaluation before? Isn't this standard in DNNs?\
> **Answer:** We would like to remark that the batch-wise treatment of the queries is in fact novel for DPPLS. In fact both DeepProbLog and NeurASP treat queries individually.
>
> **16.** gradient for the solution does not have to be positive: zero? This should not be in the appendix.\
> **Answer:** We would like to remark the following about the value of the gradient vector $\tfrac{\partial P_{\Pi(\theta)}(Q)}{\partial p}$. We consider the example of $1 + 1 = 2$ for MNIST Addition. Since there is only one possible solution of the equation, the gradient vector will consist of one positive entry and nine negatives, i.e., $[-, +, -, -, -, -, -, -, -, -]$. The meaning of the positive entry is to reward the right solution in the probability vector of, $P_{\theta}^{(C|X)}(x)$ and negative ones to penalize the wrong ones. In total, gradient of values of zero will not appear. The values of the gradient of zero do not occur because zero says that the concrete solution is neither wrong nor right. This is the direct contradiction of our work.
>
> **17.** Attention vs slash attention: you define the latter in page 8?\
> **Answer:** Indeed, we defined SLASH Attention in page 8 of our paper and would like to recite its definition in the following form. SLASH Attention's computational graph is subdivided into two parts: Slot Attention Encoder and four PCs, each learning the representation of discrete properties, i.e., color, shape, shade, and size. Further, we refer to Figure 1 as the exact depiction of its learning process and Appendix B for its source code.
>
> **18.** Please refer to the sw you use in the text (eg, which ASP solver?)\
> **Answer:** Assuming you mean software with 'sw', we have implemented SLASH in Python using the following software packages: CLINGO - a grounder and solver for logic programs, Pathos library for parallelization of ASP-calls and PyTorch for the Deep Probabilistic Models standing behind NPPs.
>
> **19.** The one flaw in this paper is that you only compare with DeepProbLog, You drop NeurASP after the first table In general. it is hard for me to fully accept your claims :( \
> **Answer:** We would like to explain our intention behind this decision. Throughout our experiments, we want to show the importance of PCs in LP. Thus, utilizing DeepProbLog in Table 1(b) as the only baseline, we have demonstrated the rich features of PCs as NPPs, since a DNN would require additional techniques to handle missing data. PCs are therefore syntactical sugar of SLASH in this regard.

---

> ### Author Response · Authors · 2021-11-15
> **Answer to the questions raised by reviewer qUza - part I**
>
> We thank you for your time and remarks. Below you can find our answers to the mentioned issues:
>
> **1.** Finally, the whole? thing is trained using grad ascent\
> **Answer:** Please accept out apologies for the somewhat misleading formulations. Yes, the gradient ascent is used to train NPPs and SLASH, but the overall training is performed by coordinate descent. Please find the detailed information on pages, 5-6 of the main text and on 13, Appendix A on the definition of the losses and their gradients.
>
> **2.** the initial probabilities were obtained from?\
> **Answer:** We apologize for the somewhat misleading formulation. We assume throughout the paper our queries being true and following the i.i.d assumption.
>
> **3.**  $\sum \text{Models} = 1$? $P(M1\text{ and }M2)$ ? - we understand as does the sum over all possible solution sum to $1$\
> **Answer:** Indeed, the aim of the training is to achieve that the probability of the query as stated in definition 3 sums over the probability of the possible solutions (see definition 2) up to one. However, at the beginning of the training, this is not the case, since NPP(s) involved procure(s) uniform distribution upon initialization.
>
> **4.** Is it correct to use the word query for a model?\
> **Answer:** We would like to clarify the differences between the terms we involve in our writings. We refer to $I$ as a possible solution to the query $Q$ in SLASH. In ASP literature, such a solution being referenced to as "stable model".  Likewise, we apologize for the confusion caused by the remark on this matter in our paper.
>
> **5.** What is the point of the LP? It is not very well motivated.\
> **Answer:** "Logic programming = Prolog", is the common association which many people have. And, although, Prolog is Turing complete and has high computational power, it comes with syntactic limitations and lack of reasoning power, thus restricting its application to problem-solving and knowledge representation. This is the case due to Prolog's Horn clause logic. We cite the work of Chiaki Sakama and Katsumi Inoue "Answer Set Programming", Journal of Japanese Society for Artificial Intelligence 25(3), pp. 368--378, 2010, for further arguments.
>
> **6.** It would be nice to have examples showing how to write it?\
> **Answer:** Please find the full versions of the SLASH programs we have used in our experiments, since Figure 1 and Figure 2(b) entail only limited versions of the SLASH Attention program. We hope these examples should be sufficient for giving initial ideas of how to write such programs.
>
> **7.** ASP; do you use non-strat neg? Agg? Everyting in your programs?\
> **Answer:** We are sorry, but we do not quite understand what you mean by non-strat neg. Could you possibly specify this in more detail? However, one can use aggregation functions, optimization and all other features CLINGO has to offer within SLASH.
>
> **8.** what goes in the NN is the solver output as a number? Ie, you run the solver every time?\
> **Answer:** The communication from SLASH to NPP(s) succeeds via our novel loss function. Yes, you are absolutely right since the solver is being called to answer each of the queries, what happens in parallel within SLASH since we are treating queries batch-wise.
>
> **9.** User vs training loss: I got lost, what is the point of a loss function that is not used in training? TensorFlow and friends allow to define a second function on tuning data, is that it?\
> **Answer:** Maybe there is a misunderstanding here, but we don't quite understand what you mean with user loss. Could you specify this further? The loss function, which we specified in equation (10) in the new version of the paper, is uniform for SLASH. That is, it is the same for all SLASH programs. Thus, we use it for all our experiments.
>
> **10.** Evaluation can you give short info on dataset size at the beginning?\
> **Answer:** The size of the data set is always determined by the queries in SLASH. These are attached to the source code in Appendix B.
> In MNIST Addition, there are always two images involved in one question. Consequently, the training dataset was 30000 in total size and the test dataset of 5000. For ShapeWorld4, one query was asked for each shape, resulting in maximally 80000 questions for training (20000 images) and maximally 20000 (5000 images) for testing in total. We have included this information in our new version.
>
> **11.** slash(pc) and slash(dnn) have the same ASP?\
> **Answer:** Maybe there is a misunderstanding. Please accept our apologies. No matter of the NPP's form, DNN+Softmax, PC, DNN+PC, a SLASH program has the same ASP together with its functionality like optimization, aggregation functions and so on.

---

### Author Response · Authors · 2021-11-15
**General comment**

We would like to remark that SLASH represents a faithful extension of DeepProbLog. If we compile a PC into a computational graph as EiNet do, then we can directly recognize an EiNet as neural predicate in DeepProbLog. However, we would have to change the semantics as we do not have a single argument of the predicate/atom to capture the output. For this purpose, we introduce NPPs to achieve a novel framework. Furthermore, we uploaded the new version of our paper.

---

> ### Author Response · Authors · 2021-11-22
> **Addition to the general comment**
>
> The new version of our paper entails the following changes compared with the initial submission:
> * p.5, **Section 3.1** — added the loss function for NPPs, $L_{NPP}$.
> * p.6, **Section 3.3** — rewritten to the greater part. Particularly, we introduced the new loss functions: $L_{ENT}$ and $L_{SLASH}$. Further, we based the overall training procedure on coordinated descent.
> * p.7, **Section 4**, **Evaluation 1** and **Table 1(a)** — deleted discussion on running times (RTE) and partially reworded the drawn bottom lines.
> * p.9, **Summary of all Empirical Results** — added one sentence.
> * pp. 13-14, **Appendix A** — entails the step-by-step derivation of  the loss functions gradients.
> * p. 18, **Appendix D** — added to discuss the results of the MNIST Addition experiments performed in the "opposite" setting, i.e., the training performed the full dataset and the test dataset entailing various rates of missing data.

---

### Decision · Program_Chairs · 2022-01-20

**Decision:**

Reject

**Comment:**

The most positive reviewers have not decided to step forward to champion the paper. Others have a negative impression which has not sufficiently changed after the answers from authors. Actually, it is acknowledge that there have been many modifications, but they are not happy enough with this situation: modifications (some significant ones) cannot always be fully checked again and even with the efforts that were made by reviewers, strong concerns remained. It has been pointed out that the direction has potential. My recommendation is based on the data that I have available.